



# Sea Ice Drift and Arch Formation in the Robeson Channel Using Daily Coverage of Sentinel-1 SAR Data During the 2016 – 2017 Freezing Season

Mohammed E. Shokr[1], Zihan Wang[2], Tingting Liu[2]

[1]Metreological Research Division, Environment and Climate Change Canada, Toronto, Ontario, Canada, M3H-5Tu
[2]Chinese Antarctic Center of Surveying and Mapping, Wuhan University, Wuhan, 430079, China

*Correspondence*: Tingting Liu (ttliu23@whu.edu.cn)

**Abstract.** Robeson Channel (RC) is a narrow sea water passage between Greenland and Ellesmere Island in the Arctic. It is a pathway of sea ice from the central Arctic and out to the Baffin Bay. This paper uses a set of daily Synthetic Aperture Radar (SAR) images from Sentinel-1A/1B, acquired between September 2016 and April 2017, to study kinematics of individual ice floes as they approach then drift through the RC. Tracking of 39 selected floes was visually performed in the image sequence and their speed was calculated and linked to the reanalysis 10 m wind from ERA5. Results show that drift of ice floes is remarkably slow while in the compact ice regime upstream of the RC unless the floe is surrounded by water or thin ice. In this case the wind has more influence on the drift. On the other hand, ice floe drift is found to be about 4–5 times faster in the open drift regime within the RC and clearly influenced by wind. A linear trend is found between change in wind and change in ice drift speed components, both along the length of the channel. Case studies are presented to reveal the role of wind on ice floe drift in details. The study also addresses the development of the ice arch at the entry of the channel. It started development on 24 January and matured on 1 February 2017. Details of the formation process, using the sequential SAR images, are presented. The arch's shape continued to adjust by rupturing ice pieces at locations of cracks under the influence of northerly wind (hence the contour keeps displacing northward). The study highlights the advantage of using the high-resolution daily SAR coverage in monitoring aspects of sea ice cover in narrow water passages where the ice cover is highly dynamic. The information will be particularly interesting for possible applications of SAR constellation systems.

## 1 Introduction

One of the exit gates for sea ice flux from the Arctic Basin to southern latitudes is through Robeson Channel (RC). It is located between Greenland and Ellesmere Island (Canada), with its northern location around 82° N, 62° W (Fig. 1). It connects the Lincoln Sea (a southern section of the Arctic Ocean) to the Kennedy Channel, which opens to the Kane Basin. These three water bodies are known as Nares Strait, which opens south to the Baffin Bay. RC is a short and narrow passage (about 80 km in length and 30 km wide) and more than 400 m deep along its axis.

Oceanographic measurements in the RC are not commonly performed. Herlinveaux (1971) found the dominant surface current to be from north to south with an average velocity in April and May increased from about 0.36 km h$^{-1}$ near the surface to nearly 0.9 km h$^{-1}$ at 80 m depth. Strong southerly current around 1.08 km h$^{-1}$ was also measured in the western section of the channel during early spring of 1971 and 1972 with fluctuation of 0.43 (Godin, 1979). When they used two



ocean simulations to study the circulation and transport within Nares Strait, Shroyer et al. (2015) found that the mean current
structure in south of RC depended on the existing of landfast ice.

The sea ice cover in the RC comprises a combination of seasonal (first year ice, FYI) and perennial ice (multi-year ice, MYI)
both imported from the Arctic Basin through the Lincoln Sea. The only locally grown ice is found in narrow strips adjacent
to the land at the two sides of the channel. Based on earlier results of ice thickness and motion retrieved from optical satellite
sensors and reconnaissance flights in the 1970s, Tang et al. (2004) estimated the ice flux crossing the RC to be around
$40 \times 10^3$ km². Using a record of ice displacement retrieved from Radarsat-1 images during 1996–2002, Kwok (2005) found
the average annual ice area flux to be $33 \times 10^3$ km². Rasmussen et al. (2010) modelled the sea ice in the Nares Strait using a
three-dimensional coupled ocean (HYCOM) and sea ice model (CICE). Their results showed a much lower ice flux in 2006
(20 km³ year⁻¹) than in 2007 (120 km³ year⁻¹) leading by blocking of the ice flow in spring of 2006.

Sea ice drift is influenced by wind forcing, ocean current and internal stresses within the pack ice (caused by interactions
between ice floes, which reduce ice momentum). The latter factor is determined by ice types and concentration within the
pack. Other minor factors include the Coriolis force and sea surface tilt. The dynamics of the ice motion is based on spatial
and temporal scales (McNutt and Overland, 2003), namely individual floe (< 1 km), multiple-floe (2–10 km for up to 2 days),
aggregate floe (10–75 km with 1–3 d time scale), pack ice cover (75–300 km at 3–7 d) and sub-basin scale (300–700 km at
7 – 30 d). The best coupling with wind occurs at the pack ice scale (also called coherent scale). According to this
categorization, the only individual and multiple floe scales are observed in SAR images of the RC. Here, the response to
wind is usually floe-to-floe bumping, ridging, redistribution and differential floe motion (McNutt and Overland, 2003).

55  Tracking individual ice floe motion from a sequence of satellite images is potentially feasible if the temporal resolution of
the satellite coverage is reasonable (at least daily). An early attempt is reported in Sameleson et al. (2006) for ice in the
Nares Strait using the coarse-resolution satellite data (tens of kilometres) from a passive microwave radiometer at 6.5 GHz.
The authors tracked the motion using only 3 to 5 locations of same floe in a sequence of the satellite images. Due to their
fine resolution (tens of meters). sequential Synthetic Aperture Radar (SAR) images are the best tool to monitor sea ice
60  kinematics, particularly if available at short-time scale. However, they have a limited spatial coverage. The earliest studies to
estimate sea ice displacement using SAR is presented in Hall and Rothrock (1981) and Leberl et al. (1983), using sequential
Seasat SAR images. Later, making use of the more frequent coverage of Radarsat-1 in the western Arctic, the Radar
Geophysical Processing System (RGPS) was developed and produced gridded ice motion and deformation data, tracked
every 3 – 6 days from 1998 to 2008 (Kwok and Cunningham, 2002). A more recent ice tracking operational system (also
65  gridded) is described in Demchev et al. (2017) using a series of Sentinel-1 SAR images.



While RC is covered most of the year by influx of ice from the Lincoln Sea, it is possible that the flow of the ice may be blocked in winter at the entrance of the channel by formation of an arch-shaped ice configuration that spans a transect between two land constriction points at Greenland and Ellesmere Islands. The arch usually collapses in early summer, allowing continuation of the ice flux. Kwok et al. (2010) pointed out that no arch was formed in 2007 leading to a major loss of Arctic ice, which was equivalent to about 10 % of the average annual amount of ice discharged through the much wider Fram Strait (400 km versus 30 km width). This signifies the fact that the entire Nares Strait can represent a major route to the Arctic if the ice arch ceases to form in the future due to thinning of Arctic ice. Moore and McNeil (2018) addressed the collapse of this arch in relation to the recent trend of sea ice thinning. In the present data set, the arch formation started on 24 January 2017 and continued till May 2017. The study includes detailed description of the mechanism of arch's formation.

The objective of the study is to utilize the daily Sentinel-1A/-1B SAR coverage of the RC area during a full freezing ice season (September 2016 to end of April 2017) to examine two sea ice features in the RC. The first is the drift of individual ice floes in terms of speed and direction. The information is linked to the 10 m wind reanalysis data to explore the wind influence on the ice drift. Further knowledge about how wind and ice drift are related enables the improvement of sea-ice/atmosphere dynamic models (Leppäranta, 2011). The second is monitoring the formation of the ice arch at the inlet of the RC during its 10 days of development until maturity. The advantage of using the daily coverage of the fine resolution SAR in retrieving this information in such a narrow channel is expected to instigate further operational applications of SAR constellation systems (e.g. the recent Canadian Radarsat Constellation Mission (RCM)) with their finer temporal resolution.

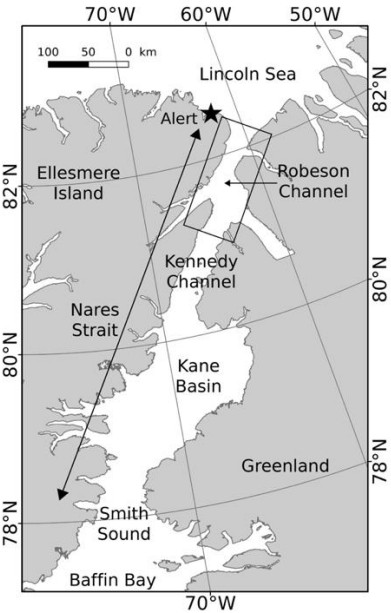

**Figure 1.** Map of the RC and its surrounding areas. It is located between Greenland and Ellesmere Island (Canada), with its northern location around 82° N, 62° W. It connects the Lincoln Sea to the Kennedy Channel, which opens to the Kane Basin.



## 2 Data Sets

### 2.1 Satellite data

Sentinel-1A and 1B are two satellites developed within the satellite constellation of the European Space Agency's (ESA) Copernicus program. They were launched on 3 April 2014 and 25 April, 2016, respectively. Both carry a carbon-copy C-band SAR (central frequency of 5.405 GHz) with a selection of single or dual polarization. Image acquisition is performed in one of four operation modes: Stripmap (SM), Interferometric Wide swath (IW), Extra-Wide swath (EW), and Wave (WV). Both IW (swath 250 km at spatial resolution 5m×20m) and EW (swath 400 km at median resolution 20m×40m) modes, both

with Level-1 Ground Range Detected (GRD) product were used. All images were acquired in HH polarization. Almost daily coverage of the RC area from both satellites were obtained from late September 2016 to the end of April 2017 (total of 361 images). Images were calibrated to backscatter coefficient in decibel then georeferenced. In order to reduce the image size and the speckle, images were resampled to 50×50 m. While the incidence angle of the EW mode varies between 29.1° and 46.0° across the swath, no correction for the variation of the angle was performed since the backscatter was not used

quantitatively.

### 2.2 Wind data

Reanalysis of 10 m level wind is available from a few sources. Four sources were examined in this study: (1) the U.S. National Centers for Environmental Prediction, jointly with National Center for Atmospheric Research (NCEP/NCAR) (Kalnay et al., 1996), (2) the joint NCEP with the Department of Energy (NCEP/DOE) (Kanamitsu et al., 2002), (3) the

European Centre for Medium-Range Weather Forecasts (ECMWF) Re-Analysis (ERA-Interim) (Dee et al., 2011), and (4) its successor ERA5 (C3S, 2017). Specifics of each source are presented in Table 1. The difference between the estimated speed from each source and the speed from Alert station is plotted in Fig. 2 for the period from 1 October 2016 to 30 April 2017. Alert weather station (Canada) is located at (82.52° N, 62.28° W). This location is close enough to the study area (Fig. 1).

Figure 2 reveals the overestimation of the reanalysis wind when the station's wind measurement is < 10 km h$^{-1}$. As the wind measured from the station increases, a systematic underestimation of the reanalysis wind is observed. This is particularly true from the two ERA products. When the speed from the Alert station exceeds 30 km h$^{-1}$, reanalysis wind from all sources can be severely underestimated by 20–40 km h$^{-1}$. Previous studies show that low-resolution global reanalysis of the wind speed and direction have large errors in the narrow channels of the Nares Strait (Dumont et al., 2009). The present data show that

the average absolute deviation of the NCEP/NCAR, NCEP/DOE, ERA-Interim and ERA5 wind reanalysis from the measured wind at Alert station is 9.12, 9.74, 9.04 and 8.92 km h$^{-1}$ over the period from 1 October 2016 to 30 April 30 2017, respectively. Hence, we chose ERA5 data because of its minimum deviation from Alert's data and finer grid spacing. Data are used to explore links with ice floe drift and study the ice arch development. The grid points from ERA5 reanalysis





relevant to the study area are shown in Fig. 3. Data from appropriate number and locations of grid points are introduced later

in the relevant sections.

**Table 1.** Information about available wind data at the study site from the Alert Weather Station and four sources of reanalysis data.

| Data | Alert Stn. | NCEP/NCAR | NCEP/DOE | ERA-Interim | ERA5 |
|---|---|---|---|---|---|
| Grid spacing | - | 2.5°×2.5° | 1.88°×1.90° | 0.75°×0.75° | 0.25°×0.25° |
| Temporal res. (h) | 1 | 6 | 6 | 6 | 1 |
| Level (m) | 10 | 10 | 10 | 10 | 10 |
| Nearest grid pt. to Alert Stn.(km) | - | 37.6 | 37.6 | 1.9 | 1.9 |

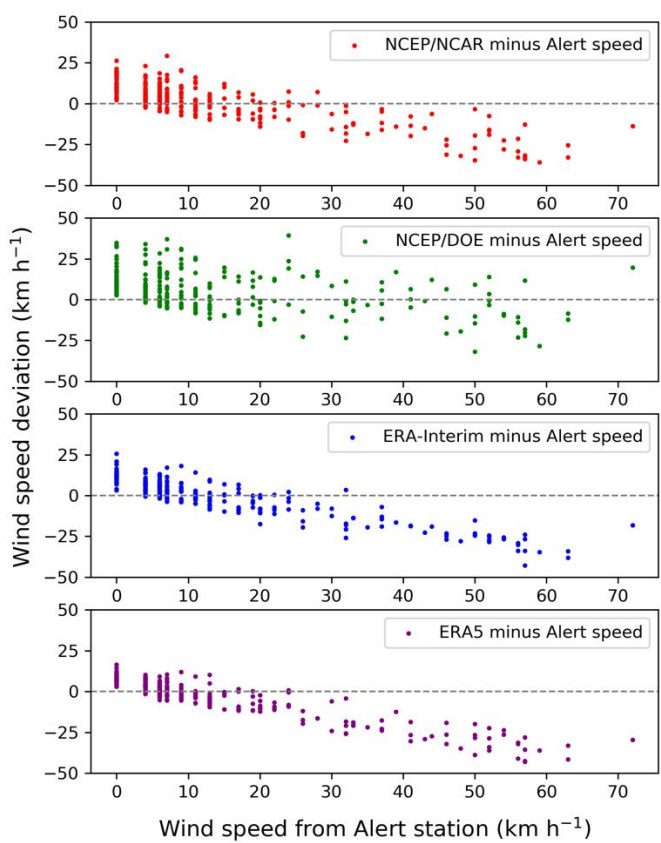



**Figure 2.** Deviation of reanalysis wind speed from the speed measured at Alert Weather Station (reanalysis minus station wind) for the period 1 October 2016 to 30 April 2017. The reanalysis data are from NCEP/NCAR, NCEP/DOE, ERA-Interim and ERA5. Note the increasing underestimation of the reanalysis data as the measured wind increases.

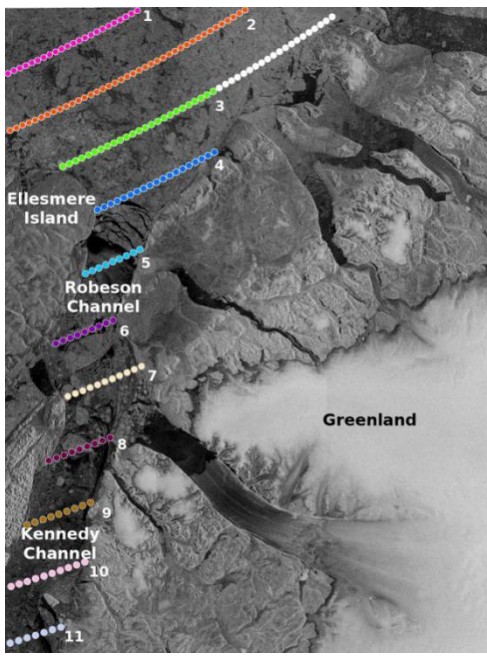

**Figure 3.** ERA5 grid points of wind reanalysis data used in the present study. The background is Sentinel-1B image acquired on 30 January 2017.

### 2.3 Ocean current and sea surface height data

Daily and monthly mean maps of ocean current (vertical coverage at 50 levels from -5500 m depth to surface) and sea surface height are components of the GLORY12V1 reanalysis product covering the altimetry era (1993–2018). It is based on the real-time ocean reanalysis product of Copernicus Marine Environment Monitoring Service (CMEMS) (Fernandez and Lellouche, 2018). Sample maps of both parameters are used in this study to compliment the interpretation of the wind influence.

### 3 Method

Daily coverage of the fine resolution images of Sentinel-1A/1B SAR (50 m) were used to generate tracking of selected sea ice floes and detect the temporal evolution of the ice arch from its commencement on 24 January until it matured on 1 February 2017. A total of thirty-nine floes were tracked between 26 September 2016 and 10 April 2017. Thirty-two floes moved mainly southward, crossing the inlet of RC, and seven moved mainly northward within the RC. Among the seven floes, five moved in the drifted ice regime before and shortly after the formation of the ice arch and two moved in a polynya-



like regime after the formation of the arch. Each ice floe was visually identified in a sequence of daily SAR images (between 4 to 28 scenes), then the floe displacement, drift speed, and direction were calculated. The displacement was determined from the subjectively estimated locations of the same floe in two successive daily images using a code to covert latitude/longitude pairs to distance according to World Geodetic System (WGS84) coordinate system.


Three sources of error are implied in the estimation of the displacement. The first is the geolocation error of the SAR image. The Sentinel-1 product specification (Bourbigot et al., 2016) mentions that the absolute pixel location accuracy is less than 7 m for the IW mode but no number is mentioned for the for EW mode, which is used in this study. The second is the assumption of a linear path (as opposed to curvilinear or meandering path) of the floe between two successive days. This

assumption had to be employed because the temporal resolution of Sentinel-1A/1B is not finer than one day. The third is the subjective estimate of the centre of the same floe in successive images. This was also estimated to be within a few pixels. Assuming these errors are independent and normally distributed, the error in the estimated ice drift speed would be roughly 0.2 km d$^{-1}$.

The ice floe speed was calculated using the travelled distance as mentioned above and the period between the two image acquisitions, which varied between 16 and 33 hours. To link the floe speed to the wind at any location of the floe, the reanalysis wind data from ERA5 were acquired from the four grid points closest to the floe. This is to avoid inclusion of wind data, which are usually highly variable, from grid points far from the relevant floe location. The 3-hour wind vectors that acted on the given ice floe during its transition during the period of the two satellite passes are produced in the form of

polar maps to qualitatively explore their influence on the floe drift. In addition, the 3-hour wind speed from the four grid points around each floe at each location were averaged to quantitatively link it to the drift speed and generate statistics to quantify the wind influence.

The evolution of the ice arch during its formation period from 24 January to 1 February 2017 was manually delineated in

each image. The daily displacement of its two end points was calculated using their latitude/longitude coordinates. For investigating the role of the wind on the progress of the arch shape as well as the location and displacement of its terminal points, daily wind data from ERA5 reanalysis were averaged from the 3-hour interval at the coloured grid points from lines 3, 4 and 5 in Fig. 3.

## 4 Results

Two subsections are presented here. The first addresses motion of selected ice floes with exploration of wind influence on the motion. For that purpose, two ice regimes: upstream and within the RC are addressed separately as their ice cover, hence the floe drift pattern, varies. The influence of the wind differs between the two regimes as explained later. Case studies are





presented for floes in each regime to reveal quantitative and qualitative information about the wind influence. The second subsection addresses the evolution of the ice arch at the inlet of the RC, highlighting the role of the wind on its formation.

**4.1 Ice floe motion**

**4.1.1 Tracking ice floe drift**

According to Arctic Council (2001), large-scale motion of the pack ice in the Nares Strait is triggered by ocean current and wind, which both push the ice from the Lincoln Sea southward to the RC. The pack encompasses ice floes of different ages and sizes. Typical dimensions of the examined floes in this study range between 2 to 16 km. Some floes are composite, i.e.
aggregates of smaller floes, which were disintegrated during their journey.

Figure 4 shows tracks of 16 ice floes heading mostly southward yet with a few interruptions to this dominant direction. Dates of each location of a given floe are marked. The thirty-nine floes are arbitrarily labelled for purpose of identification. Nonetheless, the numbers have no significance to the present analysis. In the convergent path upstream of the RC (with its
end marked by the strait line in the top left panel in Fig. 4) the high concentration of ice floes (compacted ice regime) slows the motion of individual floes and induces their meandering paths (in less concentration regime the floe would be more influenced by wind as shown later). Once the ice floe crosses the bottle neck at the entrance to the RC, it becomes released from the stresses induced by surrounding ice. Thus, the speed increases remarkably and the drift direction follows the north-south extension of the channel. This is mostly consistent with the current direction and the dominant wind direction (mostly
northerly though it can be reversed occasionally). More qualitative and quantitative data to reveal the role of wind and ice concentration on ice floe motion are presented in Sect. 4.1.3. Figure 4 shows also that ice floes never entered any fjord at any side of the channel. In fact, many fjords become filled by locally grown landfast ice early in the freezing season.





**Figure 4.** Track of 16 selected ice floes, obtained from the daily Sentinel-1 images, as they approach and pass through the RC. Note the slow motion upstream the channel and the faster motion through the channel. The entrance to the channel is marked by the solid line in the top left panel.



### 4.1.2 Ice floe drift speed

Figure 5 shows the average drifting speed of each ice floe (regardless of drift direction) during its entire observation time in the SAR time series; either upstream or within the RC. Upstream the RC, the speed varied within a narrow range (4–10 km d$^{-1}$), with a typical value around 5 km d$^{-1}$. Such a nearly constant drift speeds, observed under different wind speeds, suggests that the wind has minor or no influence on the floe drift. The exceptionally higher average speed of floe #6 (~ 19 km d$^{-1}$) resulted from floe drifted in a surrounding area of nearly zero ice concentration that prevailed for the three days. A similar example is presented in the case study 1 in Sect. 4.1.3.

The situation is different for the floes drifted within the RC. Here, the floe speed became remarkably higher, typically between 14 and 45 km d$^{-1}$ (yet, one floe reached an extreme value around 99 km d$^{-1}$ on one day as shown in the case study 3 below). The higher drifting speed inside the RC is partly explained by the low ice concentration (as observed in SAR images) and/or the prevalence of thin ice, particularly observed after the ice arch matured on 2 February 2017. Both give rise to a more significant influence of wind on ice drift. The large variability of floe speed (which contrasts the nearly constant speed upstream RC) is attributed to the influence of the variable wind speed and direction. For example, the remarkably high average speed of ice floe #4 inside the RC (45 km d$^{-1}$ as shown in Fig. 5), as reflected by the large leap of locations during the period 10–13 November (Fig. 4), is instigated by a dominant northerly wind between 20 and 40 km h$^{-1}$ during that period. On the other hand, the relatively slow drift speed of floe #5 (15 km d$^{-1}$ as shown in Fig. 5), demonstrated also by its limited distance travelled between 18 and 26 October (Fig. 4), resulted from reversed southerly wind between 10–20 km h$^{-1}$ during 220 same period. The effect of the adverse (i.e. southerly) wind neutralizes the action of southward current. The combined effects of wind and current on the ice drift along the channel direction are presented in the next section.

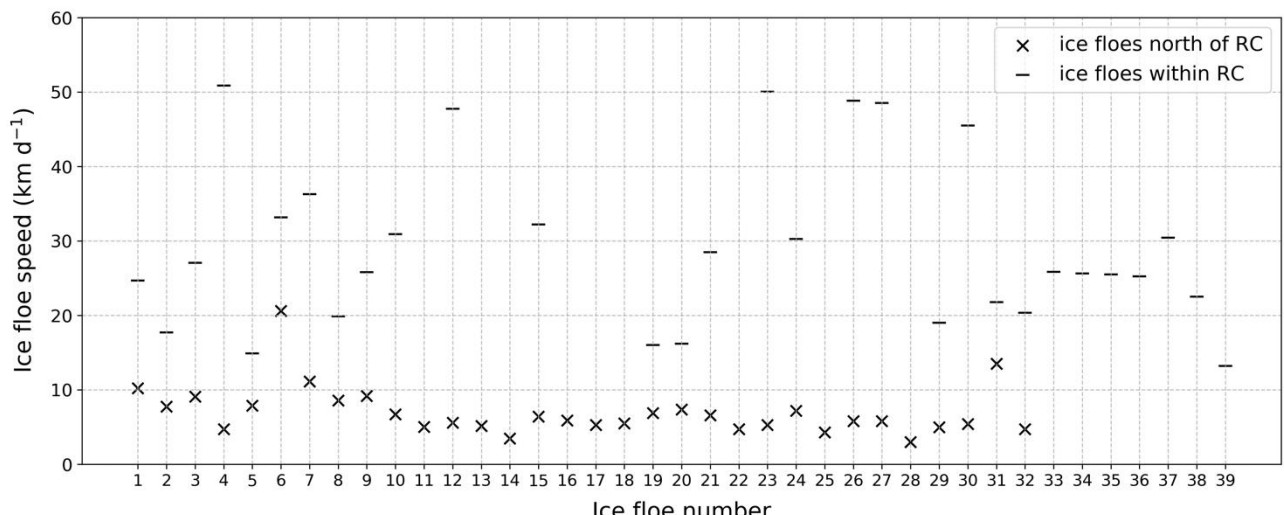

**Figure 5.** Average speed of individual ice floes during its life period upstream and within the RC. The last seven floes (#33–39) floated within the RC only with highly variable speeds.



### 4.1.3 Driving forces of ice floe motion

As mentioned in Sect. 1, ice dynamics at the floe and multiple-floe scales may not be strongly coupled with wind forcing. It is rather triggered by a combination of wind, ocean current, internal stress within the pack ice, Coriolis force and sea surface tilt. While several studies have addressed the influence of wind forcing on the large-scale motion of the pack ice (which are typically coherent), results presented in this section address the influence of the wind, given the associated ice conditions, on ice drift. Isolating wind from other influences is a challenging task, particularly within a closed pack ice such as the one encountered upstream of the RC. An attempt to achieve this task using a modelling approach is presented in Thorndike and Colony (1982) and Kimura and Wakatsuchi (2000). As mentioned above, links between wind and ice floe drift were explored in two ice regimes; upstream and within the RC separately.

### *1) Ice drift upstream of RC*

Synoptic information of the ice regime north of the RC is presented using the six selected scenes shown in Fig. 6. A bulge-shaped area of consolidated ice appears to be formed attached to the coast of Greenland. It is delineated by the dotted line in all images except in the image of 26 October, though it is still barely visible in this image. This can possibly be a large extent of landfast ice though it was exposed to cracking as appears in the image of 7 November. On 5 November, an offshore southeasterly wind ($8 - 16$ km h$^{-1}$) possibly instigated the cracking. This continued on 6 November, but on 7 November onshore wind of $15 - 30$ km h$^{-1}$ closed it. An arch-like crack is visible in the image of 13 November with its boundary coincided with the bottom boundary of the landfast ice area. This was probably instigated by the strong northerly wind (20–60 km h$^{-1}$) which prevailed on 8 November. This arch, however, did not survive because its west-side terminal (left side in the image) was not anchored on land (more on this point is presented in the next section).

Figure 6 shows that ice which entered the RC follows a path coming around the northern section of Ellesmere Island as shown by the arrow in the image of 1 December. No ice appears to enter of the RC coming along the coast of Greenland. The pack ice motion is driven by both the strong northerly winds, which is channelled down the atmospheric pressure gradient from the Lincoln Sea to Baffin Bay (Gudmandsen, 2000) and the ocean current. Additionally, it might be driven by differences in the sea surface height (SSH). Sample maps of surface ocean current (monthly averaged from December 2016) and SSH are presented in Fig. 7. Ocean current is remarkably small north of RC yet it accelerates at the inlet and within the channel. On the other hand, the SSH exhibits an obvious west-east gradient around the tip of Ellesmere Island. There is also a gradient in the same direction within the RC. These two factors govern the large-scale motion of the pack ice. The erratic drift of ice floes within the pack north of the RC (Fig. 4) nullifies any suggested influence of ocean current or SSH. Ice floe motion upstream of the RC appears to be mainly determined by interactions between neighbouring floes in the closed pack ice. This observation is illustrated in the following case study.





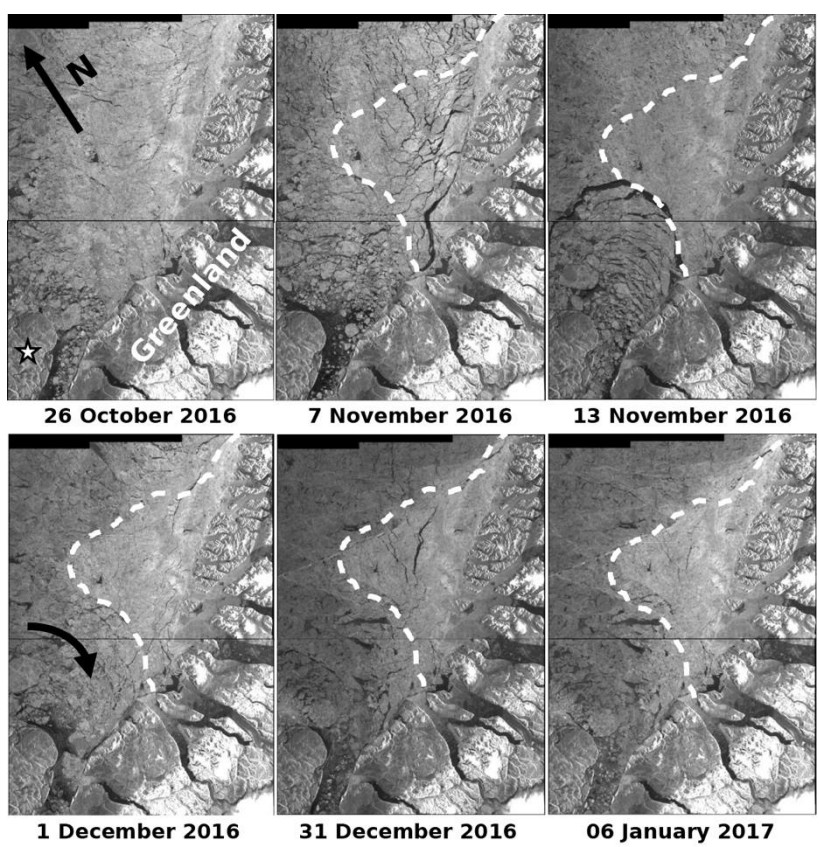

**Figure 6.** Sequence of Sentinel-1A/1B images of an area north of RC. The dotted curve marks an area of consolidated ice (still visible in the 26 October image). Ice cracked in this area on 7 November and an ice arc was formed on 13 November. The star marks Ellesmere Island. Ice floes that made their way to the RC originate from the west (not north) following the bath shown by the arrow included in the 1 December image.

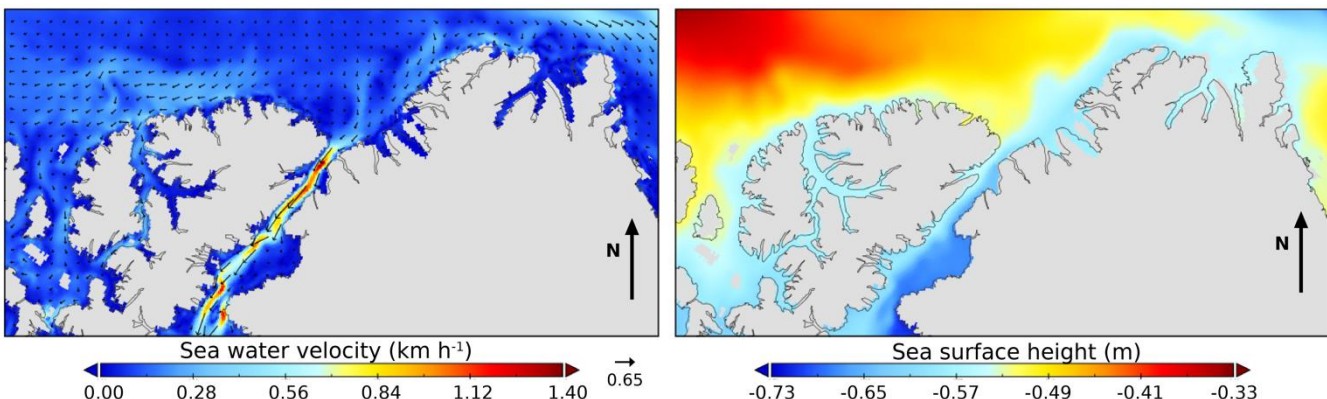

**Figure 7**. Maps of ocean current near the surface (left) and sea surface height (right), averaged over December 2016 from GLORYS12V1 which is generated by Copernicus Marine Environment Monitoring Service.



***Case study 1: two floes drifted upstream of the RC***

Figure 8 shows sequential Sentinel-1 images (26 September to 7 October 2016) of a segment just upstream of RC where two ice floes appear; floe #2 is marked by the grey dot (natural low backscatter area) and floe #3 is marked by the star. The corresponding maps of the 3-hour ERA5 reanalysis wind vectors from the closest four grid points surrounding the two floes
at each location between the two successive images are presented in Fig. 9 (same grid points for both floes). Speeds of each floe during the period between two successive image acquisitions are listed in Table 2. This information assists in defining the impact of the wind on the floe drift as explained below.

The image of 26 September shows the two floes surrounded by open water and thin ice (more so than for floe #3). The wind
between the two satellites overpasses on 26 and 27 September (averaged 33 km h$^{-1}$) was partly heading northeast or southeast. This relatively high wind, combined with the less resistive ice in the surrounding, caused floe #3 to drift northeast at highest speed of 24 km d$^{-1}$ (Table 2) and reached its location on 27 September (Fig. 8). Between 27 and 28 September, relatively light wind (<20 km h$^{-1}$) blew in opposite directions but floe #3 drifted southeast at 18.05 km d$^{-1}$ because this path (through open water) was less resistive. Floe #2 did not move far (drift speed was only 2.84 km d$^{-1}$) as it was surrounded by
ice. Between 28 and 29 September, the light wind did not change and the two floes remained at same location. When the wind direction changed between 29 and 30 September to head southeast at speed reaching 30 km h$^{-1}$ the two floes drifted in same direction with more notable motion of floe #2 (at speed 15.72 km d$^{-1}$ as shown in Table 2). Here, once again, the path was ice-free. During the short period between 30 September and 1 October when south-easterly wind blew at near 30 km h$^{-1}$, ice drift accelerated to nearly 9 km d$^{-1}$. After 1 October the wind abated but the floe drift continued in southeast direction at a
moderate speed between 2–6 km d$^{-1}$. When wind blew northward again between 4–7 October (exceeded 40 km h$^{-1}$ during the first 3 days then became <30 km h$^{-1}$), floe drift did not follow the wind action in the first two days as the two floes were surrounded by high ice concentration. Nevertheless, northeasterly drift is observed between 4-5 October, particularly of floe #2 (Fig. 8), following the strong south-easterly wind during the same period (Fig. 9). This case study demonstrates the effective role of wind on ice floe drift when surrounded by thin ice or water along the wind direction.




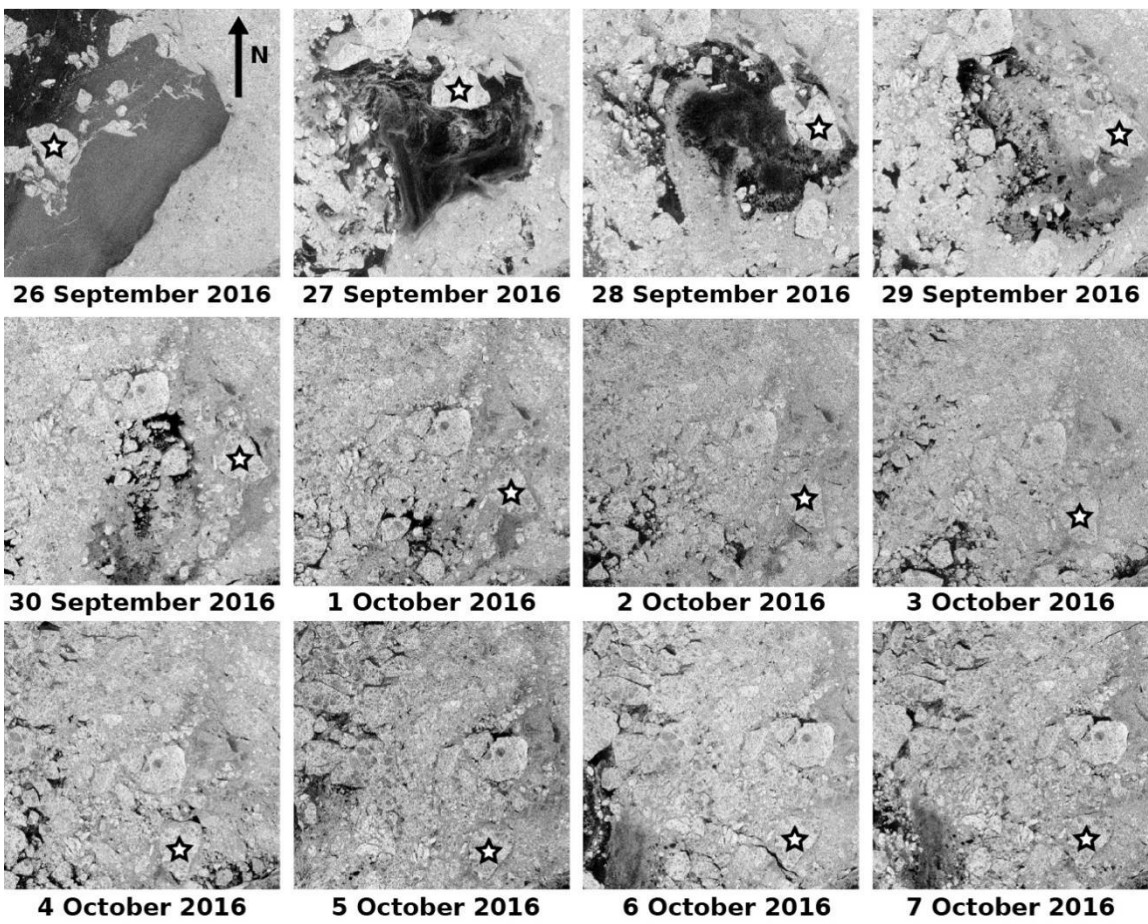

**Figure 8.** Sequential Sentinel-1A/1B images (dates are shown) showing advancement of two ice floes; floe #2 is marked by a grey dot (a natural low backscatter area), and floe #3 is marked by a star. The ice concentration surrounding each floe is visible and can be qualitatively estimated.



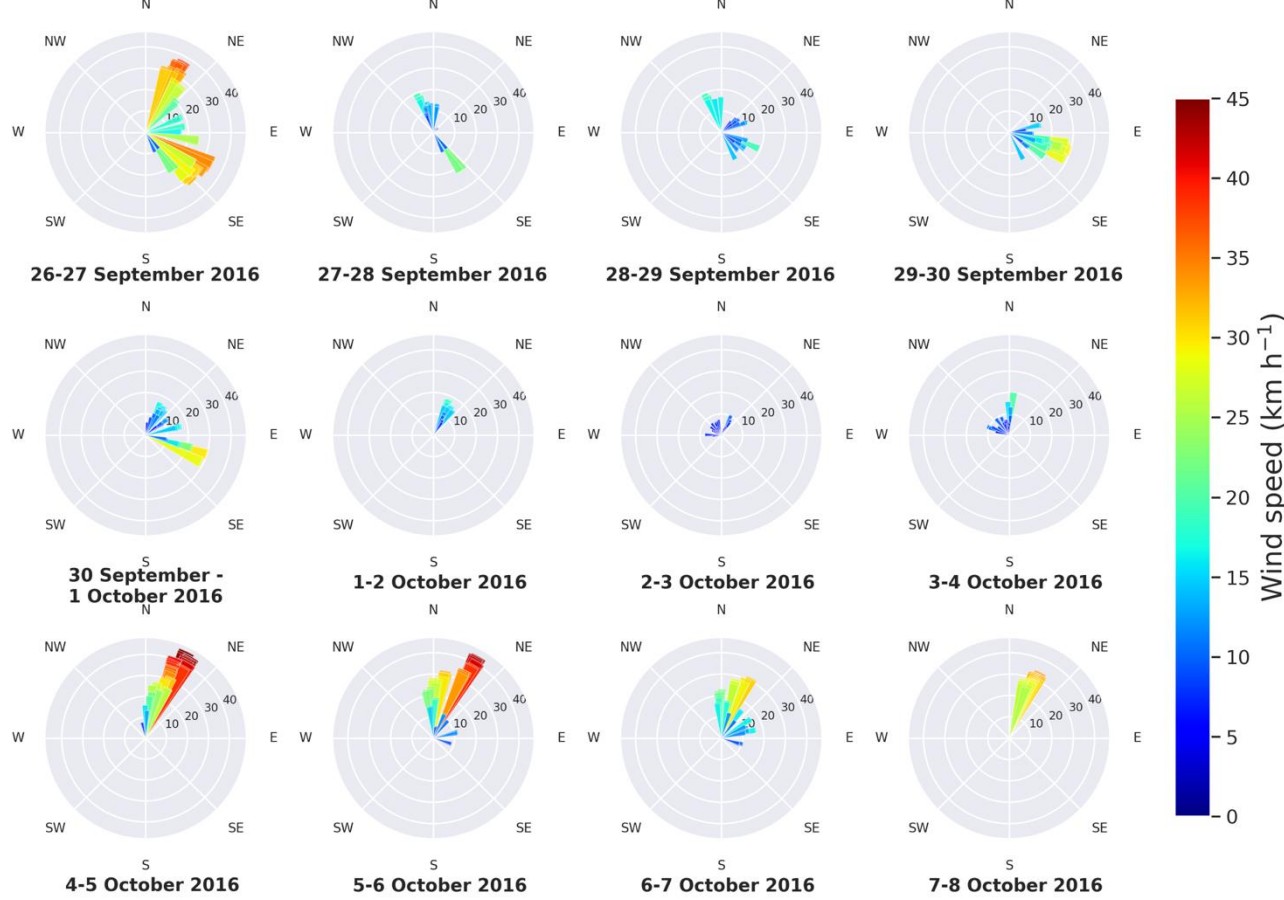

**Figure 9.** Maps of 10 m level wind vector (km h$^{-1}$) from ERA5 reanalysis. Each panel has vectors from the four grid points surrounding the location of ice floes #2 and #3 every 3 hours during the time between the daily overpasses of Sentinel-1. For example, the 26−27 September panel has the 3-hour vectors between the satellite acquisition times of 26 and 27 September.

**Table 2.** Drift speed of ice floes #2 and #3 (shown in Fig. 8) during period between acquisitions of two successive Sentinel-1 coverage. The period is shown in the first column.

| Date | Speed (km) | |
|---|---|---|
| | Floe#2 | Floe #3 |
| 26–27 September | 9.68 | 24.17 |
| 27–28 September | 2.84 | 18.05 |
| 28–29 September | 3.97 | 3.32 |
| 29–30 September | 15.72 | 5.26 |
| 30 September–1 October | 9.73 | 8.85 |
| 1–2 October | 6.59 | 3.72 |
| 2–3 October | 2.62 | 4.59 |
| 3–4 October | 6.26 | 4.58 |
| 4 - 5 October | 10.46 | 4.39 |





| 5 - 6 October | 5.37 | 5.15 |
| 6 - 7 October | 1.21 | 0.94 |
| 7 - 8 October | 7.91 | 9.99 |

## 2) ice drift within RC

Two features make ice drift inside the RC different from the drift upstream of the RC. The first relates to the ice cover and the second is associated with the ocean current. Prior to the formation of the ice arch on 24 January 2017 (see next section), the channel was filled with ice floes transported from the north but the ice concentration was moderate. Such regime is called "drifted ice", a term normally used when ice concentration is less than 60 %. This contrasts the term "pack ice" used when the concentration exceeds 70 % as observed upstream of the RC. Once an ice floe crosses the inlet to the RC, it becomes partially relieved from the stresses induced by adjacent floes, hence it becomes more responsive to wind and surface current influences. Shortly after the full development of the ice arch, the channel becomes covered with thin ice and open water, typical of polynya cover. Ice floes then become more responsive to wind and current forcing (i.e. free from effects of internal stresses).

The second feature is about the dominant southward current in the channel (Fig. 7) that reaches speeds between 0.72 and 1.38 km h$^{-1}$. This would be powerful enough to influence the floe drift either before or after the arch formation. Although daily and hourly ocean current data in the RC are available from GLORYS12V1 and PSY4 (Lellouche, 2018), it was not possible to explore the relative weights of wind and current forcing on ice drift. An ice dynamic model would be more suitable for this purpose. Nevertheless, the influence of the wind is described qualitatively below with few quantitative data.

Within the RC, ice floes advance along the channel's direction (heading mostly southward but occasionally northward as shown in the track of floe #29 in Fig. 4). Such nearly linear path made it easy to explore links between drift and wind speed. This was performed by considering their speed components along the channel's length. Scatter plots of these two components are presented in Fig. 10. Positive values indicate motion northward and vice versa. It can be seen that southward-blowing wind (i.e. northerly wind) is always associated with southward ice drift. The situation is different when the wind blows northward. In this case, ice floes may remain drifting southward (influenced by the current). However, as wind accelerates, the ice may eventually drift northward. This is shown in the reversed path of floe #29 from 10 to 18 November (Fig. 4) and explained in the discussions below.

Figure 10 includes also two sets of data. The first encompasses 37 floes (marked by open circles in the figure) that entered the RC from the north and floated in the drifted ice regime that encompassed several floes. The second set includes data from two floes (marked by solid circles), which assumed their drift in the RC when it was covered with thin ice and water (a polynya) after the formation of the ice arch. One is floe #38, which separated from landfast ice in the RC. The other is floe



#39, which broke off from the ice arch on 5 March 2017 and remained in the RC until 11 April (three dates are shown in Fig.

18). The figure shows that when the wind is heading south the ice drift follows same direction. But when the wind is heading

north, the drift can be in same or opposite direction. Here, the reverse direction of the current (i.e. heading south) counteracts

the wind action. The trend in Fig. 10 is defined by the linear regression equation $fs = 1.498ws - 22.369$, where $fs$ and $ws$

are the floe and wind speed, respectively. This equation suggests that in absence of wind the ocean current would induce floe

drift (southward) at approximately 22 km d$^{-1}$. The coefficient of determination ($R^2$) of the regression equation is 0.599.

However, when only the two floes drifted in the thin ice cover (#38 and #39) are considered, $R^2$ increases to 0.680,

indicating the more influence of wind when the surface features thin sheet and water. Higher $R^2$ value means the less

variability in the data, i.e. the better the linear regression model in representing the variation.

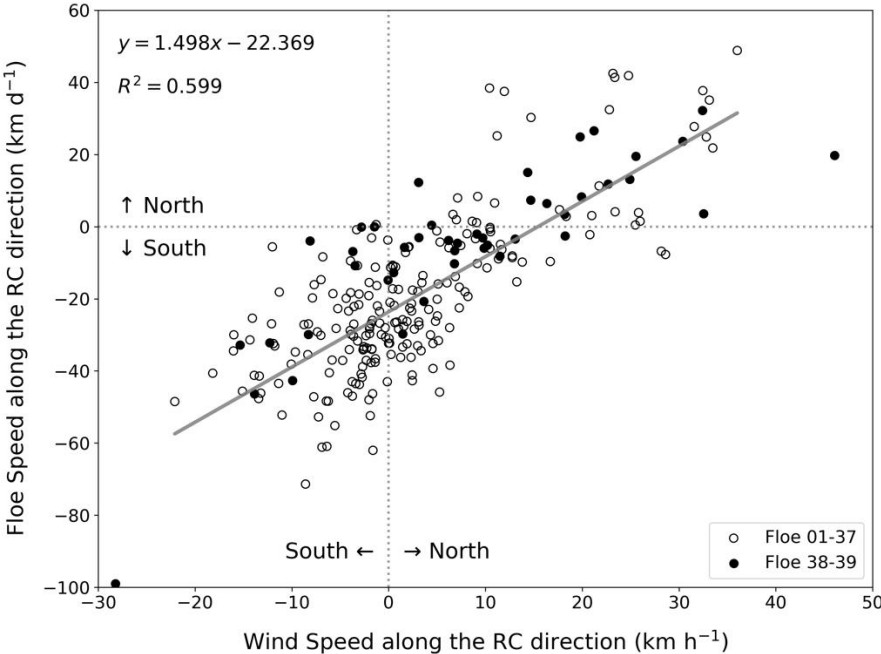

**Figure 10.** Scatter plot of wind versus ice floe speed components along the RC direction. Positive and negative values pertain to wind or
ice motion heading north or south, respectively. Data from 39 floes drifted within RC are shown. Open circles pertain to 37 ice floes
originated north of RC then floated in the drifted ice regime in the RC (with many floes existed). Closed circles represent data from 2 floes
originated inside the RC and floated in the polynya formed after the ice arch formation. The dashed line is the linear regression for the 37
floes data.

In order to assess the impact of surface wind on ice drift along the RC direction while taking into consideration the drift

direction (northward or southward), a scatter plot of the daily change of wind speed ($\Delta ws$) and floe speed ($\Delta fs$) (signed data)

was generated (Fig. 11). This is calculated as the speed (of wind or ice) in a given day minus the speed in the previous day.

The positive sign indicates northward motion. Here, the wind speed is averaged from the closest four grid points around each





floe at each location. The large scattering of the data points is attributed to the influence of factors other than wind; namely

ocean current, dynamic interaction between floes and proximity of floes to land or landfast ice. The linear regression

equation is $\Delta fs = 0.931\,\Delta ws - 1.218$, with small $R^2$ of 0.296.

Figure 11 includes three sets of data. The first set (open circles) encompasses data from 32 floes moved mostly southward in

the drifted ice regime before the arch formation. The scattering is quite high, with $R^2$ assuming a low value of 0.181. The

second set (grey circles) includes five floes drifted mostly northward in the same regime (yet with a smaller number of

interactive floes) before and shortly after the ice arch formation. $R^2$ reaches a higher value of 0.322; an indication of

enhanced wind influence. The third set includes data from the two floes (mentioned above), which drifted in the polynya

(black circles). $R^2$ increases to 0.467. This plot quantitatively confirms that wind has more influence on ice drift as the ice

regime includes a smaller number of floes and eventually a polynya-like formation.

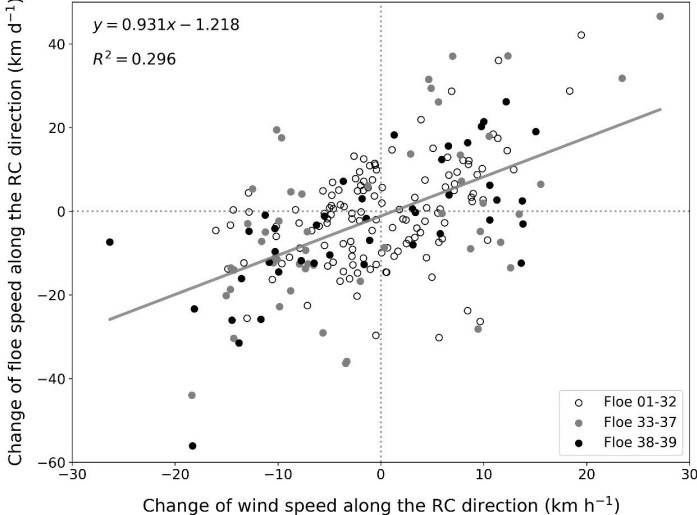


**Figure 11.** Scatter plot of daily change of wind speed $\Delta ws$ and the corresponding daily change of ice drift speed $\Delta fs$, calculated from 32
floes moved in the drifted ice regime before the arch formation (open circles), 5 floes in the same regime but with a smaller number of
floes (grey circles), and two floes originated and drifted in the polynya after the arch formation (black circles).

*Case study 2: An ice floe moving in the drifting ice regime in the RC*

Figure 12 shows a sequence of daily Sentinel-1 images from 14 to 19 November where many ice floes originated from the

north of RC are seen. The path the floe marked with the asterisk (floe #29) is linked to the coincident wind vectors in Fig. 13.

There is a remarkable displacement of the floe between 14 and 15 November at a calculated drift speed of 27.0 km d$^{-1}$. The

wind speed was between 10 and 20 km h$^{-1}$ and varied over a wide range of angles (north to south as shown in Fig. 13).

Apparently, this drift was not as much influenced by the wind as it was by the impact of the incoming floes in such a high ice



concentration regime. Between 15 and 16 November, relatively southerly wind blew at speed between 20 and 37 km h[-1] but its effect was neutralized, once again, by forces from the incoming ice flux. The entire set of floes appear to drift eastward. Recall that SSH has a gradient (climatically speaking) that matches this drift direction (Fig. 7). Between 16 and 18

November the same persistent southerly wind continues but it overcame the other effects. The entire set of floes drifted northeast following the wind. The speed of floes #29, indicated in the relevant segments, was highest between 16 and 17 November (11.77 km d[-1]). Between 18 and 19 November there was no wind but the momentum continued to drift along same direction at slower speed.

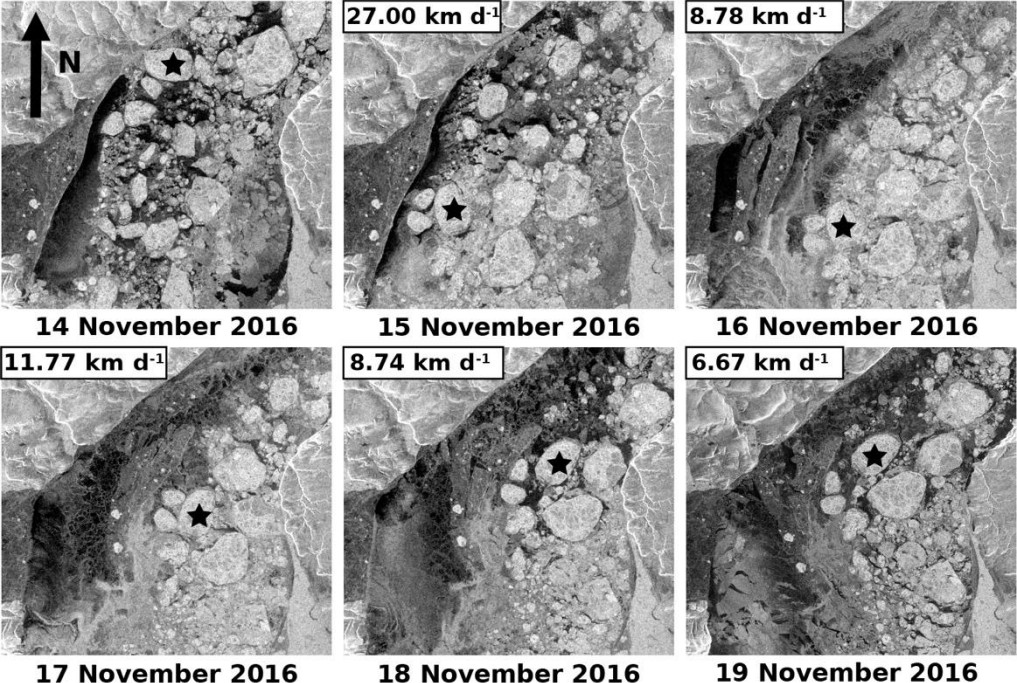

**Figure 12**. A sequence of daily Sentinel-1 images showing path of a number of ice floes. The floe marked by asterisk (floe #29) is the subject of the comments in the text. Dates of the images are shown as well as the speed of the marked floe.



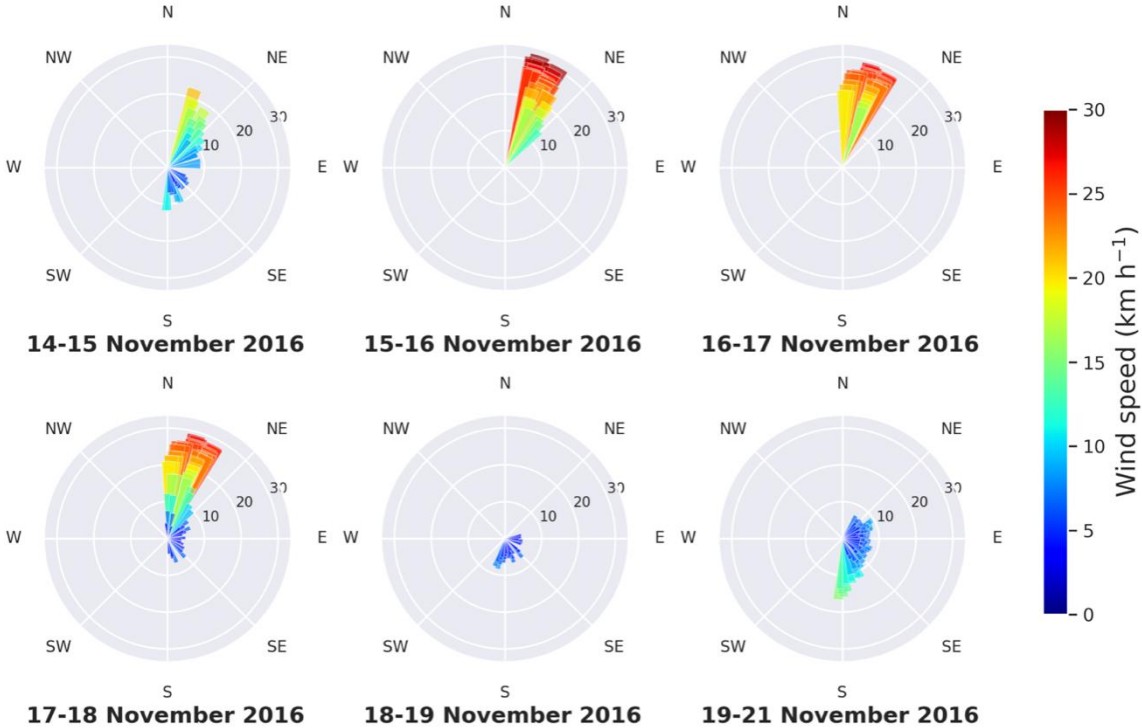

**Figure 13.** Maps of 10 m level wind vectors (km h$^{-1}$) from ERA5 reanalysis. Each panel has vectors from the four grid points surrounding the location of ice floe #29 every 3 hours during the time between the daily overpasses of Sentinel-1. For example, the 14−15 November panel has the 3-hour vectors between the acquisition times of 14 and 15 November.

*Case study 3: An ice floe drifted in the polynya within the RC*

The track of ice floe #38, which broke off from landfast ice at the Greenland side and drifted north then south in the polynya regime in the RC, is shown in Fig. 14. The track covered the period from 8 to 22 February 2017 (after the arch formation). The daily wind vector maps associated with selected floe location are presented in Fig. 15. Between 10 and 11 February southerly wind dominated though never exceeded 20 km h$^{-1}$. The floe drift (around 12 km d$^{-1}$) matched the wind direction. Between 13 and 17 February the southerly wind accelerated, reaching 40 km h$^{-1}$ then 50 km h$^{-1}$. This left an impact on the observed floe track that extended northward with floe speed reaching 11.8 km d$^{-1}$, 32.2 km d$^{-1}$ and 20.4 km d$^{-1}$ on 15, 16 and 17 February, respectively. The speed was significantly reduced to 3.7 km d$^{-1}$ on 18 February as the floe approached the ice arch (a natural barrier). After that day, the wind became northerly and the floe changed its direction of motion to advance southward. It is interesting to note the high floe speed of 43.0 km d$^{-1}$ between 20 and 21 February and the remarkably highest speed of 99.1 km d$^{-1}$ between 21 and 22 February. The latter was triggered by the highest wind encountered in this study, gusting to 50 km h$^{-1}$. However, it is important to recall that the surface current drives ice motion in the same direction. This



case study demonstrates that wind would be the prime driving force of floe motion in a regime of thin ice and combined with the current might set the drift of anomalous speed.

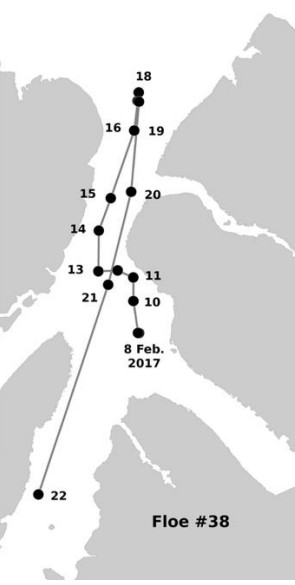

**Figure 14**. Track of an ice floe (floe #38) that separated from landfast ice and drifted in the polynya regime downstream of the ice arc. The track is shown from 8 to 22 February 2017.



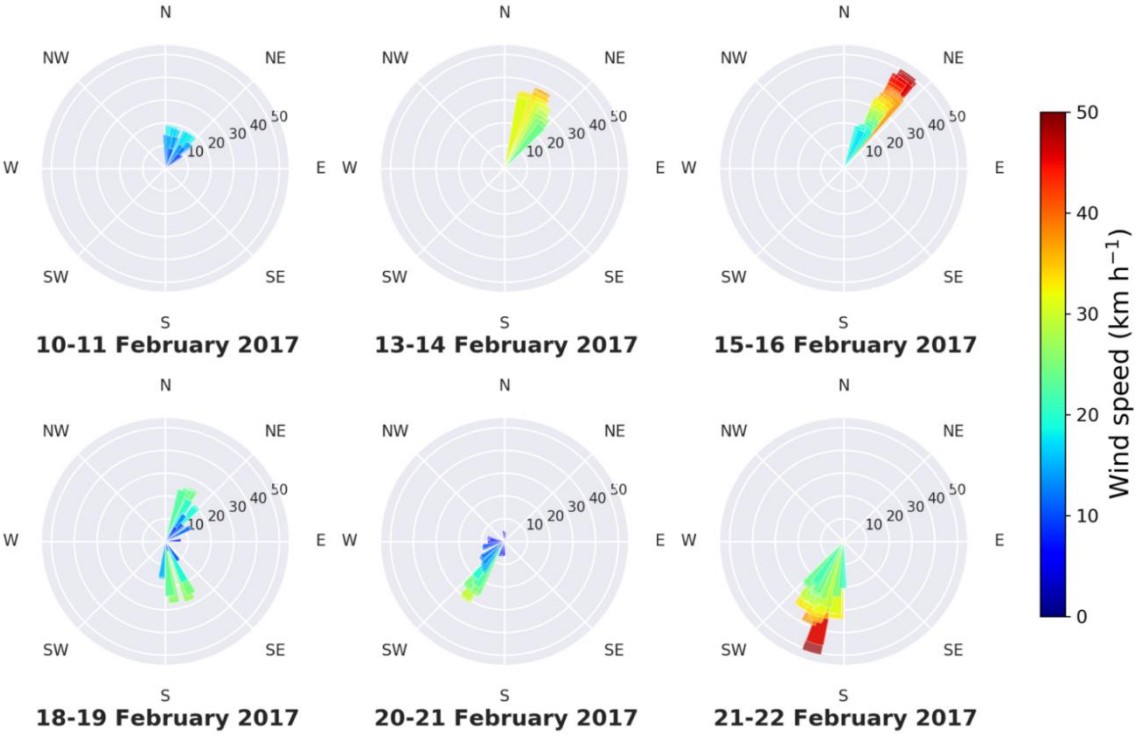

**Figure 15.** Maps of 10 m level wind (km h⁻¹), same as shown in Fig. 13 but for the ice floe #38, which was separated from landfast ice and drifted in the polynya downstream from the ice arch formed at the inlet of the RC.

## 4.2 Formation of the ice arch

The ice arch phenomenon is usually associated with the formation of a polynya downstream of the arch. Sometimes it becomes a necessary condition for polynya formation. It is well known that polynya can be driven by wind action that removes newly formed ice (latent heat polynya), and/or warm upwelling ocean water that melts the ice as soon as it is formed (sensible heat polynya) (Smith et al., 1990). However, if continuous advection of ice from a nearby source keeps feeding the area that would-be a polynya, then polynya can only be formed if a natural barrier is developed to block the advection. This obstacle would be an ice arch; a mechanically strong formation that can withstand the massive dynamic load of the advected sea ice. Obviously, this factor is irrelevant to coastal polynyas as they are backed by land (which are more common in the Antarctic region). In the case of the RC, ice arch is usually formed at the inlet of the channel, blocking the ice flux from the Lincoln Sea into the RC. It may collapse a few weeks after formation or persist as late as mid-August (Samelson et al., 2006). The arch observed in the present data set started development on 24 January matured on 1 February and collapsed in May 2017. The mechanism of arch formation is described below. After its initial formation, chunks of ice continue to detach from the arch's contour under action of northerly wind. This alters the arch's shape and the location of its



terminal points on the two constriction points at the Greenland and Ellesmere Island sides. The sequence of the development
is revealed in the set of Sentinel-1 images in Fig. 16.

The white dashed line that appear in some panels in the figure represents the arch's contour of the following day. For
example, the dashed line in the image of 24 January represents the arch's contour that appears in the image of 25 January
and so on. No line is presented if the contour remains unchanged in the following day (e.g. the cases of 25 and 26 January).
The difference between the visible arch and the dotted line in the image of any given day identifies the ice piece that was
chopped by the wind on that day. The 3-hour wind vectors during the period between two daily overpasses of Sentinel-1,
obtained from ERA5 reanalysis, are presented in Fig. 17. Data were obtained from the grid points located on lines 3, 4 and 5
in Fig. 3 and shown in the same color as appear in that figure.


After seven days of persistent southerly wind, northwesterly wind returned for a few hours between 22 and 23 January. A
wide rupture of ice cover, not arch-shaped, is observed in the 23 January image (Fig. 16). Between 23 and 24 January, the
dominant northerly wind (about 30 km h$^{-1}$) was enough to cause many cracks in the ice cover and introduce the first visible
contour of the arch (image of 24 January). On 25 January, the cracked ice drifted south and another piece of ice was
detached from the arch. On that day light southerly wind occurred (<20 km h$^{-1}$) but varied over the entire angular range.
Hence the ice drift must have been triggered by another cause (likely sea current). The same wind prevailed until 28 January
and no change in the arch was observed. With the continuation of the same wind on 29 January, numerous cracks appeared
on that day. When strong northerly wind (reaching 30 km h$^{-1}$) blew between the two satellite overpasses on 29–30 January,
the cracked ice was pushed further south, leaving a well-defined arch shape seen in the image of 30 January. A major
displacement of the arch's end point at the Ellesmere Island's side (13.88 km) is observed. The arch shape continued to be
adjusted on 31 January and 1 February in response to the same strong northerly wind. Note the two large pieces of ice that
detached in these two days (Fig. 16). After 1 February, the arch remained unchanged, regardless of the wind speed and
direction, until it collapsed on 11 May. The only exception was the breakup of a large piece, defined as floe #39, on 5 March
(Fig. 18). This piece broke while the wind was dominantly southerly (between 15 km h$^{-1}$ and 30 km h$^{-1}$). This is not
consistent with the aforementioned scenario of the modulation of the arch shape under the action of northerly wind. However,
it should be noted that the rest of ice cover in the image of 5 March (Fig. 18) appears to shift north following the southerly
wind as indicated by the arrow.

The arch legs (an engineering term that refers to the end parts of the arch) are observed to be perpendicular to the land
contour (see images of 1 and 2 February in Fig. 16). As all the forces in an arch geometry are transferred as a compression
forces, the perpendicular ending of the arch shape provides a robust way to transfer the loads directly to the rock base at both
sides. This prevents the arch from collapsing (Karnovsky, 2012). Without this configuration, the end point of the arch may




continue to slip at the surface, leading eventually to the failure of the arch. This feature, along with the curvature of the arch can be of interest to the community of ice mechanics.


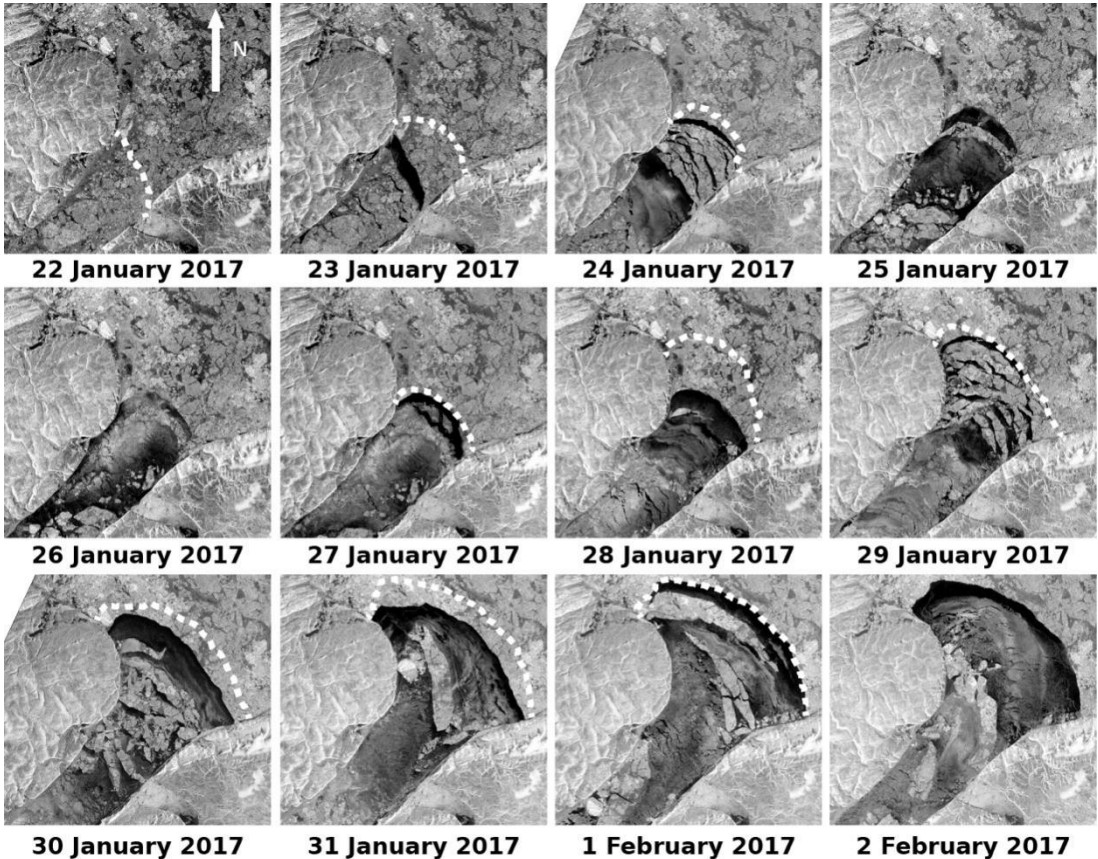

**Figure 16**. Daily Sentinel-1 images showing the development of the arch formation from 24 January 2017 matured on 2 February 2017. The dotted line marks the arch shape and location in the following day.



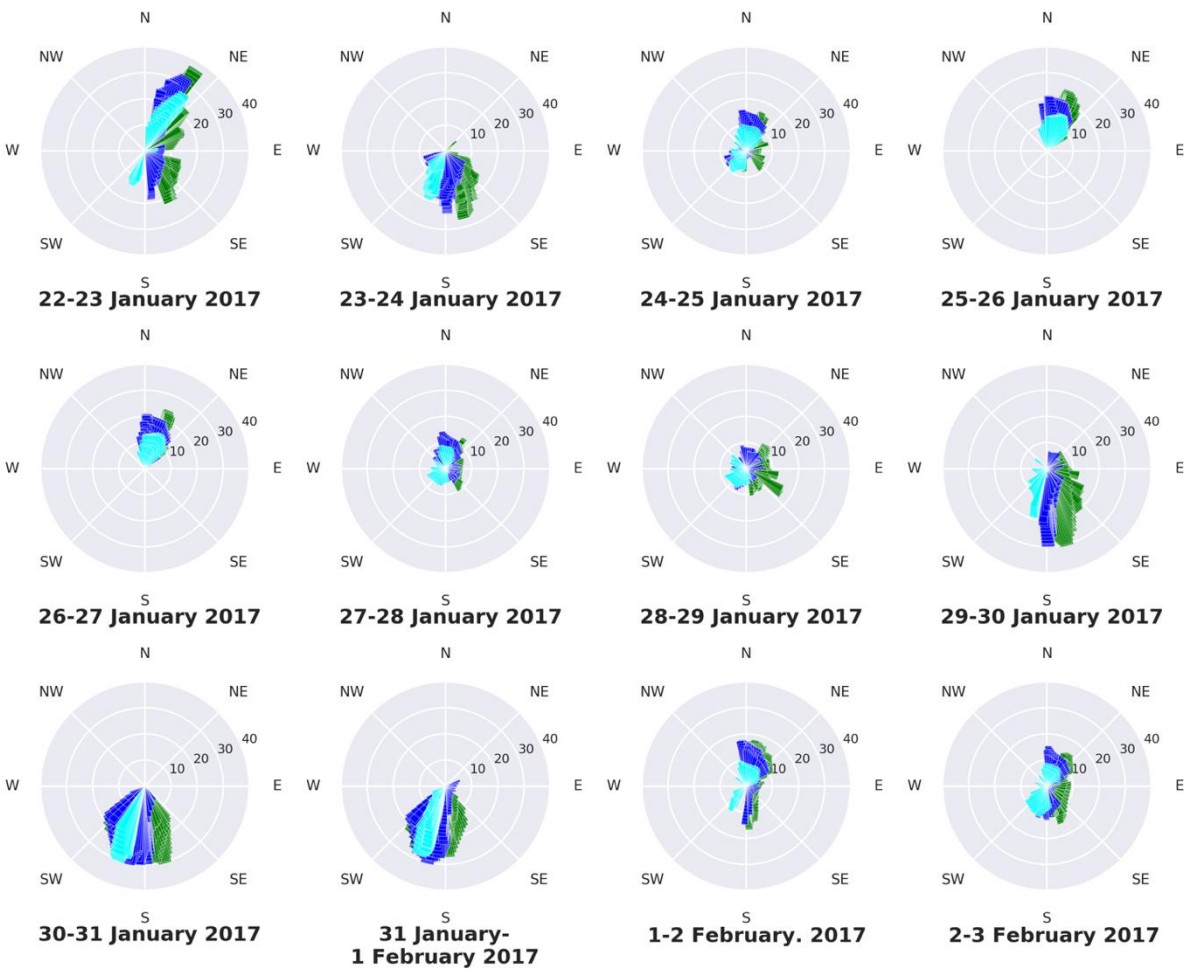


**Figure 17.** Wind speed (km h[-1]) during the formation period of the ice arch (22 January to 3 February, 2017), obtained from grid points on lines 3, 4 and 5 of ERA5 reanalysis (Fig. 3). The colors of the vectors are same as the colors of grid points in Fig. 3.

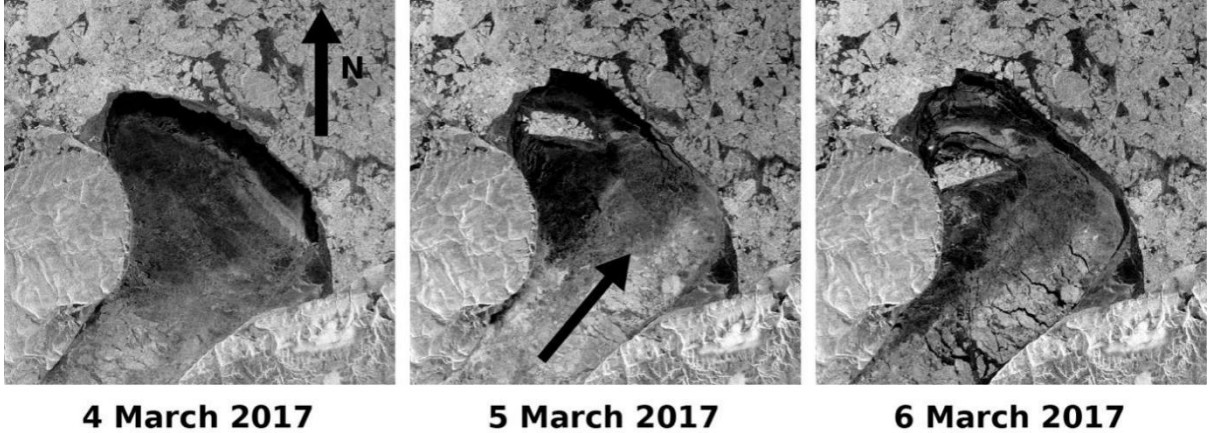



**Figure 18.** Sequence of Sentinel-1 images showing breakup and drifting of an ice piece that was labelled floe #39 in this study. This
modulated the ice arch. The arrow indicates the dominant wind direction between the acquisition times of the images of 4 and 5 March.

The above discussions highlight the mechanism of ice arch formation. Strong northerly wind plays an important role in modulating the arch's contour as it may detach pieces of ice at locations of fractures. Southerly wind, on the other hand, has virtually no effect on the arch's shape and location. In two occasions ice pieces are observed to detach in presence of light

southerly wind, which suggests a possible influence of sea surface current. The modulation of the arch's shape abates when the upstream ice becomes too compacted to allow further chopping in response to northerly wind. Moreover, the mechanically strong structure of the arch cannot fail under the dynamic force of the incoming ice flux. The structural properties of the arch are not addressed in this study except for the observed configuration of its terminal (arch legs) being perpendicular to the land surface.

**5 Conclusions and recommendations**

A series of daily Sentinel-1A/1B images have been used to study the sea ice motion at the scale of individual floes in the RC, located between Greenland and Ellesmere Island, and the process of the ice arch formation at the northern entry of the channel. The study period spanned the fall and winter seasons of 2016/2017. Wind data from ERA5 reanalysis were used to explore the role of wind on ice floe drift and the arch formation process. Thirty-nine floes were visually tracked in sequential

daily images and their speeds were calculated. Qualitative and statistical data of ice drift were obtained in two regimes, upstream and within the RC. Case studies of drift of selected ice floes in relation to the local wind are presented. The local wind was obtained from the closest grid points to the floe at each location on its track.

Sea ice that approaches the RC follows a path around the northern section of Ellesmere Island. No ice drift was observed

along the coast of Greenland in this set. The drift of the ice floes just north of the RC reveals that all floes drift at a fairly constant speed around 5 km d$^{-1}$ along erratic paths. This motion seems to be determined by internal stresses between floes with virtually no influence from wind except when the floe is surrounded by thin ice or open water. The drift speed was found to reach 15 km d$^{-1}$ in this case.

Once an ice floe crosses the entry to the RC and becomes released from the stresses engendered by the surrounding ice it starts to accelerate. While inside the channel, the floe drift speed varies between 15 km d$^{-1}$ and 45 km d$^{-1}$, following the channel's direction (mostly southward but sometimes northward, depending on the wind direction). This nearly linear motion made it easier to explore links between wind and floe drift when components along the channel's length are considered. Regression analysis between these components confirms the increasing influence of the wind on ice floe drift



when the ice cover features thin sheet and water. Change of wind vector is found to be linearly related to the corresponding change in ice floe drift within the RC with less variability in the data after the formation of the ice arch.

Monitoring the ice arch formation revealed its development over a 9 day period from 24 January to 1 February 2017. During this period the arch's shape and its terminal points continued to adjust as northerly wind keeps chopping pieces of the ice
cover upstream of the arch at locations of fractures near the arch's contour. Southerly wind has no role on this process as it closes gaps that potentially lead to ice detachment. The process continues until the pack ice upstream of the arch becomes fully consolidated and the arch takes on a mechanically strong dome shape.

This study demonstrates the possibility of generating a non-gridded ice drift vector by tracking the drift of individual ice
floes in daily SAR images. Such images are recently available, yet on a limited basis, from Sentinel-1 system, and more so from the recently launched Radarsat Constellation Mission (RCM) (a three-spacecraft fleet). More availability of data from constellation systems is expected in the future from a few national and commercial agencies. The challenge in developing this product remains is developing an automated tracker of individual floes, considering their deformation, rotation, breakup and amalgamation while drifting.

***Data availability***. Sentinel-1 data is available free of charge from the Copernicus Open Access Hub (https://scihub.copernicus.eu/dhus/#/home, European Space Agency, (last access: 2 January 2020). All reanalysis data are publicly available. ERA5 hourly data is available from https://cds.climate.copernicus.eu/cdsapp#!/dataset/reanalysis-era5-single-levels?tab=overview (last access: 20 December 2019), ERA-Interim from
https://apps.ecmwf.int/datasets/data/interim-full-daily/levtype=sfc/ (last access: 13 May 2019), NCEP/NCAR from https://www.esrl.noaa.gov/psd/data/gridded/data.ncep.reanalysis.html (last access: 13 May 2019), NCEP/DOE from https://www.esrl.noaa.gov/psd/data/gridded/data.ncep.reanalysis2.html (last access: 13 May 2019) and GLORYS12V1 from http://marine.copernicus.eu/services-portfolio/access-to-products/?option=com_csw&view=details&product_id=GLOBAL_REANALYSIS_PHY_001_030 (last access: 12
December 2019). The Alert Station data can be obtained from: https://climate.weather.gc.ca/historical_data/search_historic_data_e.html (last access: 5 May 2019).

***Author contribution***. MES prepared the manuscript, designed the experiments and performed data analysis. ZHW collected and processed the data. TTL contributed to performance of data analysis and supported writing and editing.

***Competing interests***. The authors declare that they have no conflict of interest.



***Acknowledgment.*** We are grateful to the following organizations for providing the data used in this study. European Space Agency (ESA) provided Copernicus Sentinel-1A/1B product. EMCWF provided ERA5 and ERA-Interim reanalysis products. NCEP, NCAR and DOE provided their reanalysis products. And CMEMS provided their GLORYS12V1 reanalysis product.

***Financial support.*** This work was supported in part by the National Key Research and Development Program of China (2018YFC1406102), the fund of the Key Laboratory of Global Change and Marine-Atmospheric Chemistry (GCMAC1806), the National Natural Science Foundation of China (41676179 and 41941010).

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
