# Peer review of "Sea Ice Drift and Arch Evolution in the Robeson Channel Using the Daily Coverage of Sentinel-1 SAR Data for the 2016–2017 Freezing Season"

_The Cryosphere, 2020_

## Referee Comment (RC1) · Anonymous Referee #1 · 25 Apr 2020

**General Comments**

The paper analyzes ice floe movement with SAR satellite data in Robeson Channel during the 2016-2017 freezing season. Individual floes are tracked using daily images and conclusions are drawn on the transport of ice with respect to wind, ice congestion and current. The paper is unable to separate the contributions of wind and current, while no mention is made of tidal affects. The paper also documents the formation of an ice arch in the vicinity of Robeson Channel during the 2016-2017 season. No historical context of this event is given. The paper has an overabundance of parenthetical

statements and some poor wording.

**Specific Comments**

46: No mention of tides. The strongest tides in the Canadian Arctic are in Kane Basin to the south. Tides can reverse the strong surface current in Smith Sound on a diurnal basis. What are the tidal effects in Robeson Channel?

56: An ice tracking algorithm using short-time span AVHRR imagery was developed in Nares Strait in 2000. doi.org/10.1080/07055900.2001.9649681

70: Kwok is referring to the absence of both the North Water polynya ice arch to south and the northern ice arch in the vicinity of Robeson Channel. The northern ice arch featured in this paper has occurred 17 times since 1979, and on four occasions it was the only ice arch in the Nares Strait system. The location of the northern ice arch is highly variable. In 2007 there were no ice arches. Normally, the northern ice arch forms first followed by the southern ice arch, at which point Nares Strait becomes solid ice. This paper does not put the ice arch into a historical context. doi.org/10.1038/s41598-019-56780-6

84: What is the re-visit time for RCM? Once per day is poor temporal resolution for ice tracking in this regime.

124: No units for the x-axis for Figure 2

154: How likely is it that the floe travels in a consistent straight line over a 24-hour period?

186-200: Very confusing. 16 floes are mentioned, then 39 floes, while the figure mentions 16 floes and shows 31 floes.

250: Can any useful conclusion be derived from the SSH map? Why is the December 2016 average shown?

295: The description of the wind vectors in the caption is incomprehensible.
376: Why were these floes moving so fast?

379: Are the authors suggesting the SSH gradient is the reason for the movement? Is there a reference for this?

380: First sentence makes no sense. What is 'it' and what other effects did 'it' over-come?

381: Why are the floes moving northeast under light winds from the south? Tide?

395: How is land fast ice different from sea ice with respect to physical properties. Would this change the drift rate?

425: Reference for Antarctic polynya statement?

426: As stated earlier, there is no context given for the ice arch formation.

**Technical Corrections**

35: existing?

- 176-177: Poorly written sentence.
- 187: '39' not 'thirty-nine'
- 194: 'Remarkably' has no scientific meaning. Quantify.

246: 'that' not 'which'

260: 'path' not 'bath'

270: Should not give directions for explanations. It is done quite often in the paper.

274-280: I lost count of how many parenthetical statements are in this paper. It really hinders the flow of the narrative.

300: Table 2 has a speed in 'km'. Per day?

324: Be consistent with wind direction. Northerly wind or southward blowing. The
terms vary throughout the paper. Not sure why northerly is defined here.

342: Do not use 'i.e.' in the narrative.

380: Why is 'continues' in present tense?

400: Three times 'remarkably' has been used the paper.

419: 'polynyas' Also, do not tell the reader what is well known.

482: 'chopping' is a poor term

---

## Referee Comment (RC2) · Anonymous Referee #2 · 30 Apr 2020

This study investigates the formation structure of the ice arch at the north end of Nares Strait and sea ice floe drift characteristics in Robeson Channel using a time series of sequential S1 A/B SAR images. While the presentation quality of the figures is generally quite good, I find that the analysis is mostly descriptive and qualitative and as such only really documents some observations and prepares some statistics as opposed to generating any new scientific information on the processes and their relative importance/contribution in comparison to previous studies or the larger sea ice environment of the region. I also strongly agree with Reviewer#1 in that the tidal influence on ice

floe movement is largely ignored, thus generating considerable uncertainty in the relative contribution of the other factors cited as being significant (ie. wind speed, ocean current estimates, ice consolidation/concentration, etc). I also found the manuscript to be overly long with perhaps 3-5 too many figures. Multiple grammatical issues (listed below) made for an onerous read. While the research is likely publishable somewhere, I feel that the manuscript might be better suited for another journal, perhaps Remote Sensing, since it largely fails to generate any new process understanding.

Minor Comments

L5 – 'Meteorological' is spelled incorrectly

L29 – You say RC is about 80km in length, but your scale bar and associated 'box' indicating RC in Fig 1 suggests that RC is more on the order of 150 km long. Either the 'box' in Fig 1 must be shortened or adjust the length in the text (L29).

L31 – ". . .. that increased . . ."

L34 – fluctuation of 0.43? . . . what does fluctuation mean? Standard deviation? Range? Pls use statistical terminology

L35 – instead of 'structure in south of RC depended on the existing of landfast ice' .. how about 'structure south of RC depended on the presence of landfast ice'?

L40 – instead of 'crossing the RC' .. how about 'transiting RC'?

L44 – leading? . . . how about 'caused'?

L46 – ocean currents

L47 – reduces

L48 – what do you mean by 'based on'? .. it is not clear what point you are trying to make here regarding spatial and temporal scales. Do you mean 'is assessed at a variety' of spatial and temporal scales?

L53 – it is unclear what 'redistribution' means here. Do you mean the floes collide, fracture and then produce a smaller floe size distribution? . . . or they drift and redistribute (causing re-orientation) themselves spatially? Pls clarify.

L59 - there is a period where I think you meant a comma

L74 - Begin the sentence with 'In this study, . . ."

L75 – 'The study includes a detailed . . .'

L78 – How about saying 'process mechanisms' instead of 'features' .. since ice drift is not really a ''feature".

L82 – Its unclear where '10 days' comes from? Why not just say 'development until maturity' . . . and save the '10 days' for either the Results or Discussion section.

L109 – this phrasing is odd . . . how about "This location is reasonably close in proximity and thus is the best available automated weather station to characterise the wind general wind field at a temporal resolution of one hour".

L137 – 'Sample maps' ? . . . do you mean 'Simple maps'?

L145 – ". . . into a polynya-like . . ."

L182 – is it not possible to find any other study to support the observation that large-scale motion of the pack ice .. other than an Arctic Council, 2001 report?

L188 – not why you are starting now to write out large numbers (ie. thirty-nine). Pls stay consistent here.

L200 Figure 4 caption – 'slow motion upstream of the channel'

L204 – "Upstream of the RC . . . "

L210 – "The situation is difference for the floes that drifted within . . ." Figure 5 caption . . . do not say 'floated' .. say 'drifted' instead.

L251 – "Ocean current is remarkably weak . . ." .. or "Ocean currents are remarkably weak" .. your choice.

Table 2 caption – " . . .. two successive daily Sentinel-1 SAR images" . . . also, there is no space in Floe#2 in table . . . that is there in Floe #3

L479 – "In two occasions, . . .."

L512 – how about use the word concave . . . instead of dome shape

---

## Short Comment (SC1) · 1 May 2020

The objective of the manuscript is declared: "to utilize the daily Sentinel-1A/-1B SAR coverage of the Robeson Channel area ... to examine two sea ice features. The first is the drift of individual ice floes in terms of speed and direction in relation to wind data. The second is monitoring the formation of the ice arch at the inlet of the RC during its 10 days of development until maturity".

As such, putting the results of ice floe motion or the existence of the ice arch in a

historical or geographic context is outside the scope of the study. The data used are not designed to provide such information.

The reason we did not include the tidal effect is because its frequency is twice per day while we have daily data from Sentinel-1. Hence any possible effect of tidal forces will be balanced over the daily period. Moreover, the tide will not affect the motion component along the length of the channel. Nonetheless, we will highlight this point and include brief information about the tide in the RC in a revised version.

There is quantitative information on ice floe displacement, velocity (magnitude and direction) and correlations of floe motion to wind. However, no quantitative data on the shape of the arch as it evolves. But we can include this.

The separation of the wind and current effects on floe motion cannot be addressed in this data set. We mention that this can be done using a modeling approach (not the subject of this study) and we quoted two published studies "Thorndike and Colony (1982) and Kimura and Wakatsuchi (2000)".

The new results from the study are: (1) motion of individual ice floes on daily basis, (2) the mechanism of ice arc formation. As far as we know, no similar results were published before. We will be happy to get information on published study on ice floe motion from fine-resolution SAR (not gridded ice cover) and on the effect of wind on the evolution of ice arch until it stabilizes. Previous studies estimated motion of the ice cover (not individual floes). This was done using coarse resolution (AVHRR or microwave sensors) or SAR after degrading the resolution. Ice floe motion from this study must be accurate because we traced each floe manually in the series of daily SAR image. Linking the evolution of the shape of the ice arch to wind in new. We did not see it included in any previous study.

We think the publication of this study would be timely because of the newly-launched (and future) SAR constellation missions (e.g. the Canadian RCM). The study shows that information on daily motion of individual floes can possibly be retrieved and would

be a valuable operational product if an automated method is developed to trace the floes. Daily SAR data from a constellation may be available by request only. So, results from this study can support a request to acquire daily images from RCM (for example) over an area of operational significance (e.g. parts of the Northwest Passage during summer) to support marine navigation. It is more important for marine operators to have information on the motion of individual hazardous floes than gridded motion of the ice sheet.

We appreciate the efforts of the reviewers in pointing out grammatical mistakes and wrong words. This will be corrected in a revised version.

---

## Author Comment (AC1) · 10 May 2020

The study visually identifies individual ice floes in a daily record of Sentinel-1 images from 26 September 2016 to end of April 2017, tracks their motion in relation to the 3-hour reanalysis wind from ERA5, and reports about the combined affects of wind, ice concentration and to some extent current on the floe motion. This approach makes use of the fine-resolution daily Sentinel-1 images to track the motion of individual ice floes.

[Figure]

We do not know of a similar approach published in open literature. Its advantage is to generate motion tracking of individual floes in such a narrow channel. Gridded ice trackers with their coarse resolution fails to provide the information in narrow channels. Non-gridded trackers based on Maximum Cross Correlation technique also used a nested correlation approach to reduce the computation time, hence do not make use of the fine-resolution of SAR. In addition, they perform correlation between ice "features" not necessarily ice floes.

The daily SAR coverages also allow us to monitor the evolution of the ice arch from its inception until it stabilized after 10 days, highlighting the effect of the wind. Once again, as far as we know, this is the first time the impact of the wind on the arch development is revealed. Previous studies on ice arch in this area and in neighboring areas such as NOW and Lancaster Sound focus on the existence of the arch, its causes, impacts and interannual record of formation. The present study does not address these points but it focuses on the development of the arch's shape, which is a new subject.

The reviewer raised the point that while the paper draws conclusions on "the transport of ice with respect to wind, ice congestion and current, it is unable to separate the contributions of wind and current". This is admittedly true but the separation can only be performed through a modeling approach. This is mentioned in the manuscript and a reference of such work, though in a different geographic area, is given.

In this study we have data of ice floes motion, wind field (speed and direction) and current. Wind is the prime factor that affects the ice motion. When we find that the wind does not explain the observed motion we resort to the current for explanation. One of the reported conclusions is that upstream of the channel, where ice congestion is observed, the motion does not seem to be driven by wind or current unless the ice concentration is low enough (we provided examples). So, the internal forces within the ice cover becomes the prime factor here. On the other hand, the motion within the RC (open drift ice regime) mostly follows the north-south extension of the channel, triggered by the combination of wind and current. Here the current is used to explain

situations when the floe motion does not match the wind direction.

The reviewer also raised the point that the effect of the tide on floe motion was not considered. We did not consider this effect because we use daily data while tide occurs twice per day. In addition to be difficult to quantify, tidal effect on motion along the channel is minor compared to wind effect.

Please keep in mind that the subject of the study is not about the relative weight of each factor that contributes to the ice floe motion. The focus is on the effect of the wind, which we have detailed data on. Effects of current and ice concentration were brought in when we do not find match between ice motion and wind. We listed all the factors in the fourth paragraph in the Introduction, though not all of them are actually used numerically in the study. We will add the tide to this list and will search for tidal information in the RC and try to include its impact but, once again, when wind does not explain the observed motion.

Regrading the point raised by the reviewer in "Specific Comments" about the ice tracker algorithm using AVHRR, we will look at this but the individual ice floe motion in the narrow RC cannot be obtained from coarse-resolution AVHRR tracker.

The point about putting the ice arch in historical context was not done because for one thing this is not the subject of the manuscript, moreover we do not have the data to do so.

Most of the questions/concerns raised by the reviewer in the "Specific Comments" will be addressed in a revised version. But here we would like to refer to some of those points and try to answer.

The revisit time of RCM depends on the geographic area but more importantly it is specified in the user's request of the data. One purpose of this manuscript is to show the possibility of identifying gross features of ice floe motion tracking in narrow passages that would be of interest to the marine operational community. Ice floe gates

within the Northwest Passage is an example. With results from this study we might be able to make a case for daily acquisition of RCM over one or more of these gates to study the dynamics of hazardous multi-year ice drift in different periods.

The effect of the SSH will be defined more thoroughly and references will be used. The question about "why are floes moving northwest under light wind from south? Tide?" will be addressed based on the times of image acquisition and tide. We will give brief historical information about the arch in the RC in Section 4.2 "Formation of ice arch" and remove the well-known information about the polynya formation. We will reduce the number of parenthesis.

---

## Author Comment (AC2) · 10 May 2020

In this study, daily high spatial resolution Sentinel-1 images are used creatively to observe the ice floe motion and ice arch formation process in the RC. As it is difficult to reach high accuracy tracking of individual ice floes using an automated scheme (e.g. using the familiar maximum cross correlation approach), ice floes are tracked manually and the motion vectors were calculated and related to wind maps from ERA5 reanalysis. The study offers information on ice drift in two regimes; north of the RC in a close pack/drift ice and inside RC in an open drift regime. The paper offers new information

that highlight the gross ice floe motion in the two regimes and the process of ice arch formation as triggered by the northerly wind that keeps modulating the shape until it stabilized. Conditions for stability are described in Section 4.2. The motion tracking has higher accuracy compared to the previous studies which used gridded motion mapping of the ice cover, not individual ice floes.

During the course of this study we found that most tracking methods cannot retrieve ice motion/ice drift results in narrow strait such as the Nares Strait (not presented in this manuscript). An examples of ice motion results is presented in Figure 1 where ice motion vectors (in green arrows) covered areas in Lincoln Sea. Only a few vectors were retrieved in Robeson Channel. (Figure 1 is from a study done by members of our group)

Available ice motion products covering the Robeson Channel are limited. Their spatial resolution is low (the highest is $0.083°$). We can verify their accuracy using the results from this study. Manual identification of ice floes is the most effective processing for retrieving ice motion using daily Sentinel-1 images.

In this study, a very large amount of data is generated using daily images. The analysis in the study offers qualitative and quantitative results. For example, we summarized the characteristics. Qualitative results address links between ice motion on one hand and wind/current and ice concentration on the other hand. Quantitative results show regression between ice floe speed and wind speed. The R2 of the regression equation is 0.599, which shows strong impact of wind on ice motion under the specified conditions. More detailed information is offered at Lines 320 to 368, including Figs. 10 and 11.

We downloaded some papers from TC and manuscripts from TCD. We don't find much difference in length compared to the present manuscript. However, we will shorten the manuscript and probably delete Fig. 4 or Fig. 5 or both.

We thank the reviewer for major and minor comments and we shall address them if a

[Figure]

revised version is warranted.

[Figure]

**Fig. 1.** Ice motion results using feature tracking method.

---

## Author Response (AR1)

Dear Professor **Yevgeny Aksenov and the Reviewers**

Revision of TC manuscript tc-2020-44:

*Sea Ice Drift and Arch Formation in the Robeson Channel Using Daily Coverage of Sentinel-1 SAR Data During the 2016–2017 Freezing Season*

We would like to thank the reviewers for taking time to review the manuscript and offering suggestions to improve the manuscript. We have revised our paper according to the reviewers' comments, which are summarized in the following. We also made major changes to improve the flow of the information and reduce the size of the text.

(1)  According to the comments of **Reviewer 1**, we obtained regular data of the ocean currents and addressed the separation of the contributions of wind and ocean currents to the ice drift and reported in Section 4.3.1 using a statistical approach. In addition, we studied the tidal effect on ice motion in the Robeson Channel extensively. Tidal effect could be assumed to have much smaller influence on ice drift in the Robeson Channel than the other regimes in Nares Strait according to Johnson et al. (2011). Meanwhile, we could not conclude from ice drift velocity records from IABP buoys how tide impacted ice drift in the Robeson Channel. This conclusion was reached after we calculate the Fourier spectra of the velocity record from each buoy and found no persistent peaks from data of 9 buoys.

(2)  According to the comments of **Reviewer 2**, we clarified the motivation and contribution of this study, and the advantage of using manual tracking of individual sea ice floes as opposed to generating gridded ice motion vectors. More statements were added in the manuscript. Although we studied two approaches about the tide effect in the Robeson Channel, we could not duplicate this work in our study because of limitations that have been clarified in the following response letter.

For more details, please refer to the two response letters attached below. In addition, the we corrected many mistakes in the language of the manuscript, then we use service from a professional English language editor from the Member Chartered Institute of Editing and Proofreading. We hope that you are satisfied with the revised version. Thanks once again for your time.

Best Regards,

The authors

**Response to Reviewer 1**

In the following we include the reviewer's comments and our response, item-by-item. RC refers to reviewer's comment.

**RC:** The paper analyzes ice floe movement with SAR satellite data in Robeson Channel during the 2016-2017 freezing season. Individual floes are tracked using daily images and conclusions are drawn on the transport of ice with respect to wind, ice congestion and current. The paper is unable to separate the contributions of wind and current, while no mention is made of tidal affects.

**Response:** We clarified the approach of the study in the opening of Section 4.3.1 in the revised version. We confirm that we mainly studied the impact of wind on ice floe motion but we also consider impacts of four other sources; ocean currents, tide (both in the Robeson Channel) and the sea surface height (SSH) at the wider scale of the Lincoln Sea. Those factors are considered only when wind does not explain the observed ice drift. This information provides the basis for the discussions of ice floe drift north and within the Robeson Channel in the two separate parts in the same section. This point is also added to the objective of the study at the end of the Introduction.

Thanks for your suggestion, we obtained regular data of the ocean currents and addressed the separation of the contributions of wind and currents to the ice drift in Section 4.3.1 using a statistical approach. However, we believe that separation of these two influences can better be achieved through a modelling approach as mentioned in the second paragraph in the sub-section titled "*ice drift within the Robeson Channel*" in 4.1.3.

As for the tidal effect in ice floe motion we expanded our study to cover this point. Explanation of what we did is included in our response to Specific Comments Line 46 below.

**RC:** The paper also documents the formation of an ice arch in the vicinity of Robeson Channel during the 2016-2017 season. No historical context of this event is given.

**Response:** Thanks for the comment. We chose 2016/2017 dataset because it was available from Sentinel-1. The mechanism of the arch development in this year must be typical of the other years. No historical context of the ice arch was given because we thought it was outside the scope of the study. For the ice arch, the objective of the study is to monitor the development of the arch's shape using the daily SAR data. We believe that the information is new because no daily SAR data were available for this region before the S1 constellation. In fact, this daily coverage was the motivation behind the study. As the reviewer pointed out, previous studies addressed the interannual variability of the arch's locations. We did not cover this information but we are showing, for the first time as far as we know, the daily development of the arch's shape until it stabilized and the role of the wind on altering the shape. We included three

references that address the historical context of the arch in Nares Strait in the first paragraph of Section 4.2.

**RC:** The paper has an overabundance of parenthetical statements and some poor wording.

**Response:** Thanks for your suggestion. We removed many parenthetical statements. We also corrected some wrong and poor sentences. The language of this manuscript was improved by a professional editor named Mark Ackerley from Member Chartered Institute of Editing and Proofreading.

**Specific Comments**

**RC:**46: No mention of tides. The strongest tides in the Canadian Arctic are in Kane Basin to the south. Tides can reverse the strong surface current in Smith Sound on a diurnal basis. What are the tidal effects in Robeson Channel?

**Response:** We checked the tide data in this region and found very few sets, mostly capitalized on opportunities of expeditions for other purposes. For example, Münchow et al., (2007) and Münchow and Melling (2008) measured ocean currents in the Nares Strait using mooring buoys, most of which were located at the southern end of Kennedy Channel. They found that tides impacted the dominant component of current in Nares Strait (as the reviewer mentioned). However, Münchow et al. (2007) indicated that the amplitudes and phases of tidal constituents varied substantially both along and across the strait. Meanwhile, Johnson et al. (2011) indicated that Petermann Fjord, at 81°N, is well above the critical latitude for the M2 tide (74.5°N). Since the Robeson Channel is above the latitude of Petermann Fjord, therefore, tidal effect is assumed to have much smaller influence on ice drift in the Robeson Channel than the other regimes in Nares Strait.

Gimbert et al. (2012) used Fourier spectra of the buoy velocity from the International Arctic Buoy Program (IABP) to discuss the impact of tidal effect in the Arctic basin. We checked all available buoy datasets and found 9 buoys operated through the Nares Strait during September-April from 1979 to 2016. We calculated the Fourier spectra of the velocity record from each buoy. Figure 1 is an example from 2 buoys.

[Figure]

Figure 1. the Fourier spectra of the velocity record from two buoys.

The spectra of motion from the 9 buoys are different, some with identifiable peaks and others with no peaks. Peaks may be linked to the regular frequency of the tide. So, we could not conclude from these data how tide impacted ice drift in the Robeson Channel. A few statements are added in the manuscript, from Line 269 to 282 to explain this point. Please see third paragraph in Section 4.1.3

Münchow and Falkne (2006) addressed two independent methods to estimate tidal currents. The first is a numerical model of barotropic tidal currents that predicts depth-averaged tidal currents on a 5-km horizontal grid. The second relied upon the method of least squares to minimize residual tidal variance in the data. However, we could not duplicate this work in our study because of two limitations; the first is the coarse resolution of the numerical model that is not consistent with spatial resolution of ice floes data in our study. The second is no horizontal gradients of tidal properties within the limited study area which has to be used in the Least Squares method. Therefore, we will focus on how to resolve tidal effect from sea ice velocity from high spatial resolution satellite images in a future study (please note that these statement are not included in the manuscript).

Münchow, A., Falkner, K. K., and Melling, H.: Spatial continuity of measured seawater and tracer fluxes through Nares Strait, a dynamically wide channel bordering the Canadian Archipelago, Journal of Marine Research, 65(6): 759-788, 2007.

Münchow, A. and Melling, H.. Ocean current observations from Nares Strait to the west of Greenland: Interannual to tidal variability and forcing, Journal of Marine Research, 66(6): 2008, 801-833.

Johnson, H. L., Münchow, A., Falkner, K. K., and Melling, H.: Ocean circulation and properties in Petermann Fjord, Greenland, J. Geophys. Res., 116, C01003, doi:10.1029/2010JC006519, 2011.

Gimbert, F., Marsan, D., Weiss, J., Jourdain, N. C., and Barnier, B.: Sea ice inertial oscillations in the Arctic Basin, Cryosphere, 6, 1187–1201, 2012.

Münchow, A. and Falkner, K. K.: An Observational Estimate of Volume and Freshwater Flux Leaving the Arctic Ocean through Nares Strait, Journal of Physical Oceanography, DOI: 10.1175/JPO2962.1, 2006.

**RC:** 56: An ice tracking algorithm using short-time span AVHRR imagery was developed in Nares Strait in 2000. doi.org/10.1080/07055900.2001.9649681

**Response:** We checked this paper and referred to it in the Introduction (in Line 61 in the revised manuscript). It included a limited data set from AVHRR between March and August 1998. The ice tracking was performed using the commonly-used method of maximum cross-correlation. While the temporal resolution was fine-enough (5 hours) the spatial resolution was poor (from AVHRR). Our manuscript generates drift of individual ice floes, which can only be obtained from the fine-resolution SAR data when it become available daily. Thanks to the Copernicus program that made the S1 SAR constellation data accessible for free.

**RC:** 70: Kwok is referring to the absence of both the North Water polynya ice arch to south and the northern ice arch in the vicinity of Robeson Channel. The northern ice arch featured in this paper has occurred 17 times since 1979, and on four occasions it was the only ice arch in the Nares Strait system. The location of the northern ice arch is highly variable. In 2007 there were no ice arches. Normally, the northern ice arch forms first followed by the southern ice arch, at which point Nares Strait becomes solid ice. This paper does not put the ice arch into a historical context. doi.org/10.1038/s41598-019-56780-6

**Response:** Thanks for this information which briefly highlights a historical context of the arch. We have included it in the opening of Section 4.2 (Line 463 in the revised manuscript). As mentioned above, the objective of the arch section in this manuscript is monitoring its development on daily basis, starting from the onset of formation until it stabilized. We believe that this information applies to any arch at any location in the Nares Strait system.

**RC:** 84: What is the re-visit time for RCM? Once per day is poor temporal resolution for ice tracking in this regime.

**Response:** There is no fixed temporal resolution (i.e. re-visit time) of any SAR system at any given location because the senor is not open for data acquisition throughout the entire orbit. It is open only for about 28 minutes or so in each orbit of about 100 minutes. Hence, the frequency of the coverage should be requested by the user and approved by the acquisition plan of the satellite. We were lucky to have the daily coverage of this area from S1 constellation (S1-A/B). The advantage of such frequent coverage, as demonstrated in this study, can be used to support daily acquisition of SAR from RCM in critical areas in the future. In a relevant study, Lange et al. (2019) also used airborne observations to backtrack sea ice floes and calculated the drift speed. According to Table 1 in this research, the minimum time interval is 3 days, and the maximum time interval is 10 days. The temporal resolution in our study was better (1 day).

Lange, B. A., Beckers, J. F., Casey, J. A., and Haas, C.: Airborne observations of summer thinning of multiyear sea ice originating from the Lincoln Sea. Journal of Geophysical Research: Oceans, 124. https://doi.org/10.1029/2018JC014383, 2019.

**RC:** 124: No units for the x-axis for Figure 2

**Response:** The unit for all segments is the same as the unit attached to the bottom segments (km h$^{-1}$). This is now mentioned in the caption.

**RC:** 154: How likely is it that the floe travels in a consistent straight line over a 24-hour period?

**Response:** It depends primarily on the varying wind direction and the geometry and mechanical properties of the surrounding floes. A statement to confirm this point is added in Section 3, Line 177 in the revised manuscript. However, there are no quantitative results due to the limited in-situ data.

We checked all available buoy datasets in the Arctic, and found that 12 buoys from IABP during the period from 1979 to 2016 drifted through the Nares Strait. We chose 9 buoys out of the 12 buoys consistent with our study period (September to the next April). The temporal resolution of the buoys data is 3 hours. We used the geographical coordinates of the first temporal point in two successive days from each buoy to calculate the drifting distance and call it simulated daily distance. Then, we summed the distances during the rest of the 3-hours interval in each day, and compared them to simulated daily distances. The $R^2$ is 0.945, which indicates close relationship between simulated daily distances and 3-hours-sum distances. Therefore, the assumption of a linear path of the floe between two successive days is reliable to be employed under the limited temporal resolution of Sentinel-1A/1B. The following graphs show results from this test. In order not to increase the size of the manuscript and the number of figures, we did not include the description and results from this test.

[Figure]

Figure 2. Difference between 24h interval distance and 3h interval distance.

In Fig. 2, "Distance-Least interval" is from 3h IABP data, "Distance-24h" is daily data using two records from two successive days. The difference is large if daily displacement is big.

[Figure]

Figure 3. 24h interval distance/ 3h interval distance and corresponding counts.

In Fig. 3, X-axis is 24h displacement (daily data using two records from two successive days), i.e. sum of 3h displacement (8 records in one day). Y-axis is the corresponding counts. This figure indicates that most 24h displacement is similar to sum of 3h displacement. But there is no significance after calculating T-mean.

**RC:** 186-200: Very confusing. 16 floes are mentioned, then 39 floes, while the figure mentions16 floes and shows 31 floes.

**Response:** Thanks for this comment. Yes, it was indeed confusing. We examined 39 floes and numbered them #1, #2, …, #39. The order has no significance. In Fig. 4 we show tracks of 16 floes out of the 39 (we deleted 4 of them in the new manuscript so there are 12 floes in Fig. 4 now) and we attach the floe numbers. It just happened that the 12 floes we selected are numbered 1, 2, 3, 4, 5, 20, 21, 23, 24, 26, 29, and 31. All these floes remain long enough in the channel, so the motion tracking reveal different features that are addressed in this section. The statements in Section 4.1.1 was modified to include the above explanation, starting from Line 209 in the revised manuscript.

**RC:** 250: Can any useful conclusion be derived from the SSH map? Why is the December 2016 average shown?

**Response:** As mentioned in the text, the SHH maps, obtained as weakly, show a gradient that contributes to the large-scale motion of the entire ice cover north of the Robeson Channel. SSH map presented in the manuscript, averaged over December 2016 from GLORYS12V1, is just an example. It is in the middle of the study period. We list all monthly SSH maps in the following figure (not included in the manuscript).

[Figure]

September, 2016                    October, 2016

[Figure]

Figure 4. Monthly SSH maps from September 2016 to April 2017.

**RC:** 295: The description of the wind vectors in the caption is incomprehensible.

**Response:** We believe the reviewer was referring to the caption of Fig.9. We modified this caption to make it comprehensible (Line 343 in the revised manuscript). Captions of similar figures were also modified. Here we repeat that in each panel of the wind data, the colors indicate the wind vectors that were available at 3-h interval between the 2 acquisition times of the satellite in the two successive days shown in the label.

**RC:** 376: Why were these floes moving so fast?

**Response:** Because these 2 floes originated in the polynya after the arch formation. So, they were not surrounded by ice floes that impact their motion. They move within water and thin ice. This point is mentioned in the beginning of the section titled "*Ice drift within the Robeson Channel*" from Line 351 to Line 355 in the revised manuscript.

**RC:** 379: Are the authors suggesting the SSH gradient is the reason for the movement? Is there a reference for this?

**Response:** Yes. Wekerle et al. (2013) mentioned that the SSH difference between the Arctic Ocean and Baffin Bay (Figure 7b in the following paper) not only leads to a net outflow from the Arctic Ocean, its variability also drives the variation of the CAA throughflow. We added this reference in the revised manuscript in Line 302.

Wekerle, C., Wang,Q., Danilov, S., Jung, T., and Schröter, J.: The Canadian Arctic Archipelago throughflow in a multiresolution global model: Model assessment and the driving mechanism of interannual variability, J. Geophys. Res. Oceans, 118, 4525–4541, doi:10.1002/jgrc.20330, 2013.

**RC:** 380: First sentence makes no sense. What is 'it' and what other effects did 'it' overcome?

**Response:** Thanks for your comment. Yes, the sentence was not clearly phrased. Now it reads "Between 16 and 18 November, the same strong wind, which approached 30 km h$^{-1}$, continued to blow to the northeast (Fig. 13), and the entire set of floes responded by drifting in the same direction" in Line 418 in the revised manuscript.

**RC:** 381: Why are the floes moving northeast under light winds from the south? Tide?

**Response:** Between 15 and 16 November wind between 20 and 37 km h$^{-1}$ blew from the south but the floe moved eastward. As we mentioned before, we could not confirm the role of the tide but the slope of SHH (west-east) is mentioned as a possible explanation. Between 16-18 November the entire set of ice floes moves in same direction of the wind. Between 18 and 19 November the wind virtually diminished, but a group of floes appeared to swirl clockwise. We checked the current data from 15 to 19 November and found it did not explain the motion. Therefore, we postulate that a combination of current and internal forces between floes determine the motion in absence of wind effect. This is now mentioned at the end of Case study 2, starting from Line 415 in the revised manuscript. This is a case where it is difficult to confirm the drive for the motion from the present data.

**RC:** 395: How is land fast ice different from sea ice with respect to physical properties. Would this change the drift rate?

**Response:** Landfast ice is not different than mobile ice in most of physical properties (e.g. salinity, density, dielectric and thermal properties, etc.). However, its crystalline structure may feature larger crystals with larger brine spacing. But this does not affect the radar signature. What makes a difference in the radar signature (and the physical appearance of the ice) is the surface roughness, which is not a physical property by the way. Mobile ice is usually rough and landfast ice is usually smooth, except at its edge. Hence, landfast ice usually has low

backscatter. As for the ice floe drift, a floe would slow down as it passes along the edge fast ice because of the shear action.

As for tracking the motion of floe #38, which we think has raised the question by reviewer, the drift is not affected by the origin of the floe but rather by the moderate concentration of the ice that surround the floe in addition to the wind/current combination. This is what Case Study 3 is showing. relates the drift to the wind speed and direction. The floe drifted in an open drift ice regime.

**RC:** 425: Reference for Antarctic polynya statement?

**Response:** Thanks for your suggestion. We added the paper by Nihashi et al. (2015) in Line 463 in the revised manuscript. In Nihashi et al. (2015), a map of all polynyas in the Antarctic region was presented. All are coastal polynyas.

**RC:** 426: As stated earlier, there is no context given for the ice arch formation.

**Response:** We included a brief context that starts with the sentence "In the case of the Robeson Channel, an ice arch commonly forms at the inlet of the channel, ….." in Line 463 in the revised manuscript. We added 3 references to help interested readers to obtain further historical context of this ice arch. This comes after the line the reviewer referred to.

**Technical Corrections**

**RC:** 35: existing?

**Response:** We changed the word to "presence" in Line 36 in the revised manuscript. Thanks for drawing our attention to this error.

**RC:** 176-177: Poorly written sentence.

**Response:** Yes, it was poorly written. The whole paragraph has been rewritten, now it reads "The ice floe motion is considered in two separate regimes: north of the Robeson Channel near its entrance and within the Robeson Channel….", starting from Line 197 in the revised manuscript.

**RC:** 187: '39' not 'thirty-nine'

**Response:** Thanks for your suggestion. We follow the requirement of TC:

- **Numbers**
  - For items other than units of time or measure, use words for cardinal numbers less than 10; use numerals for 10 and above (e.g. three flasks, seven trees, 6 m, 9 d, 10 desks).

We use words less than 10, use numerals for 10 and above. All the numbers in the manuscript has been checked and modified now.

**RC:** 194: 'Remarkably' has no scientific meaning. Quantify.

**Response:** Thanks for your suggestion. We quantified it "by a factor of 1.5–5" in Line 217 in the revised manuscript. This range is obtained from Fig. 5.

**RC:** 246: 'that' not 'which'

**Response:** The sentence's structure has been modified in Line 297 in the revised manuscript. Thanks.

**RC:** 260: 'path' not 'bath'

**Response:** It was corrected in Line 312 in the revised manuscript. Thanks.

**RC:** 270: Should not give directions for explanations. It is done quite often in the paper.

**Response:** Removed part of the explanation in Line 316 in the revised manuscript.

**RC:** 274-280: I lost count of how many parenthetical statements are in this paper. It really hinders the flow of the narrative.

**Response:** Thanks for your suggestion. We eliminated many of those parenthetical statements to make the reading flows better.

**RC:** 300: Table 2 has a speed in 'km'. Per day?

**Response:** Yes, km $d^{-1}$. It is now corrected in Table 2 in the revised manuscript.

**RC:** 324: Be consistent with wind direction. Northerly wind or southward blowing. The terms vary throughout the paper. Not sure why northerly is defined here.

**Response:** Northerly, southerly, northwesterly, etc. are consistent with meteorological jargon. It is shorter in writing. We changed the term to northward blowing, southward blowing, northeastward, etc.

**RC:** 342: Do not use 'i.e.' in the narrative.

**Response:** We removed "i.e." and adjusted the sentence in Line 371 in the revised manuscript.

**RC:** 380: Why is 'continues' in present tense?

**Response:** It was changed to the past tense and modified the sentence starting from Line 420 in the revised manuscript.

**RC:** 400: Three times 'remarkably' has been used the paper.

**Response:** The word "remarkably" was removed in the revised manuscript.

**RC:** 419: 'polynyas' Also, do not tell the reader what is well known.

**Response:** 'Polynyas' is corrected. Yes, indeed we are telling the reader what is well known in one sentence that addresses the two mechanisms of polynya formation (citing the original reference of Smith at el. (1990)). The reason for this is to introduce the following sentences that are telling the reader what is not well-know; namely a necessary, though not sufficient, condition for polynya formation. That is the formation of an ice arch. This is necessary in order to block a possible stream of ice drift from a nearby source as in the case of the present area, namely the ice coming from the Lincoln Sea. This ice continues to flow to fill the location where the physical conditions for the polynya formation are indeed satisfied. Without the arch, there will be no typical polynya of considerable amount of open water and thin ice even if the conditions of polynya mechanisms are satisfied. We did not find this information in the literature and we would appreciate it if someone could point it out a reference in a previous publication so we can use it.

We deleted 'It is well known that' in Line 458 in the revised manuscript.

**RC:** 482: 'chopping' is a poor term

**Response:** 'chopping' is replaced by "detachment of ice pieces" in Line 518 in the revised manuscript.

**Response to Reviewer 2**

In the following we include the reviewer's comments and our response, item-by-item. RC refers to reviewer's comment.

**RC:** This study investigates the formation structure of the ice arch at the north end of Nares Strait and sea ice floe drift characteristics in Robeson Channel (RC) using a time series of sequential S1-A/B SAR images. While the presentation quality of the figures is generally quite good, I find that the analysis is mostly descriptive and qualitative and as such only really documents some observations and prepares some statistics as opposed to generating any new scientific information on the processes and their relative importance/contribution in comparison to previous studies or the larger sea ice environment of the region.

**Response:** The study generates information and statistics of ice drift in the narrow water passage of the Robeson Channel. During the course of this study we found that most tracking methods cannot retrieve ice drift in narrow straits. And even when they cover the larger ice cover. they do not retrieve the motion of individual ice floes. Motion of hazard ice floe is an operational requirement, which yet to be developed. The daily SAR images is the only tool to achieve this purpose. The present study demonstrated this potential. Recent and future SAR constellation systems are geared towards fulfilling this requirement. Sentinel-1 system has been the first one.

An examples of ice motion vectors from an operational product is presented in the following figure using a commonly-used feature tracking method. As can be seen, most retrieved vectors are located in the Lincoln Sea with virtually no vectors in the Robeson Channel.

[Figure]

We acknowledged the availability of S1 data from the Copernicus program at no cost, which facilitated this study.

As the focus of this study is on motion of individual ice floes (not the overall ice cover), comparison to previous studies was not included because those studies focus on ice motion in the Nares Strait, with particular attention on Kennedy Channel and Kane basin. In our view, the new scientific information presented in the manuscript is about the motion of individual floes and the relative contribution of the wind and ocean current when floes are drifted in open drift regime (low ice concentration). We demonstrate that when ice floes are surrounded by high concentration the contributions of these two factors are reduced. Also, we think the information about development of the ice arch, as triggered by wind, is new and was made possible only using the daily SAR coverage.

Information from the study can be used to verify ice drift products from coarse-resolution products (the finest is at 0.083°). We are currently assessing four sea ice motion products in the RC using daily Sentinel-1 images from September 2016 to August 2017.

Admittedly, the process of estimating the drift vectors in this study starts with manual tracking of several ice floes in sequential images. While this is a subjective and laborious task, we believe it is most accurate. This is now mentioned in the last paragraph of the Introduction with more justification of the objective of the study. The analysis includes qualitative descriptions and quantitative data. In this revised version we added more quantitative analysis by combining wind and ocean current data in a multivariate analysis to asses the relative weight of each factor on floe motion (we downloaded daily gridded ocean current data in the Robeson Channel). With this statistical approach we found that wind and ocean current explain 72.9% and 16.5% of the floe motion, respectively.

We would like to assert that this study is mainly about identifying links between ice floe motion and the wind field. That is because we have regular gridded reanalysis wind data. When wind does not explain the ice floe motion, we explore other factors starting with ocean current (daily gridded data also available). This is now mentioned in the last paragraph of the Introduction. We qualify the wind contribution as "significant" when we find that floe motion follows wind direction with speed proportional to wind speed. Results show that wind contribution is important when floe moves within low ice concentration or when the surrounded ice cover constitutes thin ice (as in the case of the polynya after the arch formation).

**RC:** I also strongly agree with Reviewer#1 in that the tidal influence on ice floe movement is largely ignored, thus generating considerable uncertainty in the relative contribution of the other factors cited as being significant (i.e. wind speed, ocean current estimates, ice consolidation/concentration, etc.).

**Response:**

We though that if the tide frequency is twice per day it will not be captured by the daily data from Sentinel-1. In this revision we expanded our study to cover tidal forcing. We checked the tide data in this region and found very few sets, mostly capitalized on opportunities of expeditions for other purposes. For example, Münchow et al., (2007) and Münchow and Melling, (2008) measured ocean currents in the Nares Strait using mooring buoys, most of which were located at the southern end of Kennedy Channel. They found that tides impacted the dominant component of current in the Nares Strait. However, Münchow et al. (2007) indicated that the amplitudes and phases of tidal constituents varied substantially both along and across the strait. Meanwhile, Johnson et al. (2011) indicated that Petermann Fjord, at 81°N, is well above the critical latitude for the M2 tide (74.5°N). Since the Robeson Channel is above the latitude of Petermann Fjord, therefore, tidal effect is assumed to have much smaller influence on ice drift in the Robeson Channel than the other regimes in Nares Strait.

Gimbert et al. (2012) used Fourier spectraof the buoy velocity from the International Arctic Buoy Program (IABP) to discuss the impact of tidal effect in the Arctic basin. We checked all available buoy datasets and found 9 buoys operated through the Nares Strait during September-April from 1979 to 2016. We calculated the Fourier spectraof the velocity record from each buoy. Here is an example from 2 buoys.

[Figure]

The spectra of motion from the 9 buoys are different, some with identifiable peaks and others with no peaks. Peaks may be linked to the regular frequency of the tide. So, we could not conclude from these data how tide impacted ice drift in the Robeson Channel. The discussion is added in the revised manuscript, from Line 269 to 282.

Münchow and Falkne (2006) addressed two independent methods to estimate tidal currents. The first is a numerical model of barotropic tidal currents that predicts depth-averaged tidal currents on a 5-km horizontal grid. The second relied upon the method of least squares to minimize residual tidal variance in the data. However, we could not duplicate this work in our study because of two limitations; the first is the coarse resolution of the numerical model that is not consistent with spatial resolution of ice floes data in our study. The second is no

horizontal gradients of tidal properties within the limited study area which has to be used in the Least Squares method. Therefore, we plan to focus on how to resolve tidal effect from sea ice velocity from high spatial resolution satellite images in a future study.

Münchow, A., Falkner, K. K., and Melling, H.: Spatial continuity of measured seawater and tracer fluxes through Nares Strait, a dynamically wide channel bordering the Canadian Archipelago, Journal of Marine Research, 65(6): 759-788, 2007.

Münchow, A. and Melling, H.. Ocean current observations from Nares Strait to the west of Greenland: Interannual to tidal variability and forcing, Journal of Marine Research, 66(6): 2008, 801-833.

Johnson, H. L., Münchow, A., Falkner, K. K., and Melling, H.: Ocean circulation and properties in Petermann Fjord, Greenland, J. Geophys. Res., 116, C01003, doi:10.1029/2010JC006519, 2011.

Gimbert, F., Marsan, D., Weiss, J., Jourdain, N. C., and Barnier, B.: Sea ice inertial oscillations in the Arctic Basin, Cryosphere, 6, 1187–1201, 2012.

Münchow, A. and Falkner, K. K.: An Observational Estimate of Volume and Freshwater Flux Leaving the Arctic Ocean through Nares Strait, Journal of Physical Oceanography, DOI: 10.1175/JPO2962.1, 2006.

**RC:** I also found the manuscript to be overly long with perhaps 3-5 too many figures. Multiple grammatical issues (listed below) made for an onerous read. While the research is likely publishable somewhere, I feel that the manuscript might be better suited for another journal, perhaps Remote Sensing, since it largely fails to generate any new process understanding.

**Response:** Thanks for the comment. We downloaded some papers from TC and manuscripts from TCD and found that our manuscript is not overly long. Perhaps papers from TC look shorter because they are presented in double column. However, considering the reviewer's suggestion, we have reduced the text and deleted Fig. 11 and the relevant text. We removed 4 panels from Fig. 4 (floes #7, #8, #12 and #30). As for grammatical mistakes, we corrected what the reviewers pointed to, checked and corrected other mistakes, and finally sent the manuscript to a professional editor to help us polish the language.

We would like to suggest that the manuscript enhances understanding of two processes. The first is about the influence of wind and ocean current on drift of individual floes in relation to the ice concentration that surrounds the floe track. The high concentration with the momentum of surrounding floes overrides possible effect of any other factor. Secondly is the monitoring the ice arch formation. This information cannot be obtained from any satellite observations other than daily SAR data, which was used here for the first time.

Of course, we could have submitted this manuscript to any journal on remote sensing for ice application but we chose The Cryosphere because it is a Copernicus publication and Sentinel-1 is a Copernicus initiative through which the data used in this study were made available.

**Minor Comments**

**RC:** L5 – 'Meteorological' is spelled incorrectly

**Response:** The spelling mistake has been corrected to 'Meteorological' in Line 5 in the revised manuscript.

**RC:** L29 – You say RC is about 80km in length, but your scale bar and associated 'box' indicating RC in Fig 1 suggests that RC is more on the order of 150 km long. Either the 'box' in Fig 1 must be shortened or adjust the length in the text (L29).

**Response:** Thanks for this comment, Fig.1 has been revised and posted in Line 100 in the revised manuscript. The 'box' of the Robeson Channel is shortened according to the figure in Kwok's paper (doi:10.1029/2005GL024768, 2005).

**RC:** L31 – "... that increased…"

**Response:** "that" has been added into the sentence in Line 32 in the revised manuscript.

**RC:** L34 – fluctuation of 0.43?... what does fluctuation mean? Standard deviation? Range? Pls use statistical terminology

**Response:** It is the Standard Deviation and now it is specified in Line 34 in the revised manuscript.

**RC:** L35 – instead of 'structure in south of RC depended on the existing of landfast ice'. how about 'structure south of RC depended on the presence of landfast ice'?

**Response:** We used "presence" to replace "existing" in Line 36 in the revised version. Thanks.

**RC:** L40 – instead of 'crossing the RC' .. how about 'transiting RC'?

**Response:** Thank you for your advice. "transiting" is better so it is used in Line 41 in the revised version.

**RC:** L44 – leading? : : : how about 'caused'?

**Response:** "caused" is used instead of "leading by" in Line 45 in the revised version.

**RC:** L46 – ocean currents

**Response:** Yes. It has been modified in Line 48 in the revised manuscript.

**RC:** L47 – reduces

**Response:** The sentence is now modified to "Internal stresses, ..., reduce ice momentum.", starting from Line 48 in the revised manuscript.

**RC:** L48 – what do you mean by 'based on'? .. it is not clear what point you are trying to make here regarding spatial and temporal scales. Do you mean 'is assessed at a variety' of spatial and temporal scales?

**Response:** Yes. The sentence is now modified to "The dynamics of the ice motion can be assessed at a variety of spatial and temporal scales, ..." starting from Line 50 in the revised manuscript.

**RC:** L53 – it is unclear what 'redistribution' means here. Do you mean the floes collide, fracture and then produce a smaller floe size distribution?…or they drift and redistribute (causing re-orientation) themselves spatially? Pls clarify.

**Response:** Thanks for your suggestion. The sentence was modified to "the response to wind is usually manifested in floe-to-floe bumping, causing ridging, fracturing, floe breaking, re-orientation and differential motion". This sentence starts from Line 55 in the revised manuscript.

**RC:** L59 - there is a period where I think you meant a comma

**Response:** There is a period after " .. at 6.5 GHz." We think it is correct.

**RC:** L74 - Begin the sentence with 'In this study, : : :"

**Response:** The sentence now begins with "In this study, ..." in Line 81 in the revised version. Thanks.

**RC:** L75 – 'The study includes a detailed…'

**Response:** "a" has been added into the sentence from Line 82 in the revised manuscript.

**RC:** L78 – How about saying 'process mechanisms' instead of 'features' .. since ice drift is not really a ''feature".

**Response:** Yes. According to your suggestion, "features" is replaced by "process mechanisms" in Line 85 in the revised version.

**RC:** L82 – Its unclear where '10 days' comes from? Why not just say 'development until maturity' : : : and save the '10 days' for either the Results or Discussion section.

**Response:** Thank you for the suggestion. "10 days of" is deleted in Line 92 in the revised manuscript.

**RC:** L109 – this phrasing is odd : : : how about "This location is reasonably close in proximity and thus is the best available automated weather station to characterise the wind general wind field at a temporal resolution of one hour".

**Response:** Thanks for this better phrasing. We have modified it to "This location is reasonably close to the Robeson Channel, and is thus the best ground source to characterize the general wind field at a temporal resolution of one hour", starting from Line 124 in the revised version.

**RC:** L137 – 'Sample maps' ? : : : do you mean 'Simple maps'?

**Response:** A new sentence is introduced: "The maps were generated at a spatial resolution of $1/12 \times 1/12°$, and both parameters were used to support the interpretation of the ice floe drift when it could not be explained by wind data only". It starts from Line 157 in the revised version.

**RC:** L145 – ": : : into a polynya-like : : :"

**Response:** Thanks for the suggestion. "in" is changed to "into" in Line 165 in the revised manuscript.

**RC:** L182 – is it not possible to find any other study to support the observation that largescale motion of the pack ice .. other than an Arctic Council, 2001 report?

**Response:** In the revised version we use another reference in Line 250 in the revised manuscript:

Kwok, R., Spreen, G., and Pang, S.: Arctic sea ice circulation and drift speed: Decadal trends and ocean currents, J. Geophys. Res. Oceans, 118, 2408–2425, https://doi:10.1002/jgrc.20191, 2013.

**RC:** L188 – not why you are starting now to write out large numbers (ie. thirty-nine). Pls stay consistent here.

**Response:** Thanks for your suggestion. We follow the requirement of TC:

- **Numbers**
  - For items other than units of time or measure, use words for cardinal numbers less than 10; use numerals for 10 and above (e.g. three flasks, seven trees, 6 m, 9 d, 10 desks).

We use words less than 10, use numerals for 10 and above. All the numbers in the manuscript has been checked and modified now.

**RC:** L200 Figure 4 caption – 'slow motion upstream of the channel'

**Response:** "of" has been added into the sentence in Line 223 in the revised manuscript. Thanks.

**RC:** L204 – "Upstream of the RC…"

**Response:** "of" has been added into the sentence in Line 227 in the revised manuscript. Thanks.

**RC:** L210 – "The situation is difference for the floes that drifted within…" Figure 5 caption: : : do not say 'floated' .. say 'drifted' instead.

**Response:** Thanks. We have changed it in Line 246 in the revised manuscript.

**RC:** L251 – "Ocean current is remarkably weak…" .. or "Ocean currents are remarkably weak".. your choice.

**Response:** We change it to "... the ocean current is very weak …." in Line 300 in the revised manuscript.

**RC:** Table 2 caption – "…two successive daily Sentinel-1 SAR images" : : : also, there is no space in Floe#2 in table…that is there in Floe #3

**Response:** The space has been added in Table 2, in Line 349 in the revised manuscript.

**RC:** L479 – "In two occasions, …"

**Response:** It has been modified to "On two occasions, ..." in Line 519 in the revised manuscript.

**RC:** L512 – how about use the word concave : : : instead of dome shape

**Response:** Thanks for your suggestion. "concave" is better to describe the shape and it has been used instead of "dome" in Line 553 in the revised manuscript.

[revised manuscript text omitted]

55 (SAR) images of the Robeson Channel. In this case, the response to wind is usually manifested in floe-to-floe bumping, causing ridging, fracturing, floe breaking, re-orientation and differential motion (McNutt and Overland, 2003).

Commented [13]: To Reviewer 2: Instead of "current".

Commented [14]: To Reviewer 2: The sentence is modified for better expression.

Commented [15]: To Reviewer 2: Instead of "is based on".

Commented [16]: To Reviewer 2: Instead of "ridging, redistribution and differential floe motion", better expression.

[revised manuscript text omitted]

Commented [34]: **To Reviewer 1**: The whole paragraph has been rewritten for easier understanding.

205   ### 4.1 Ice floe motion

**4.1.1 Tracking ice floe drift**

Commented [35]: **To Reviewer 2**: The reference of Arctic Council (2001) is deleted and we added another reference in Section 4.1.3.

The ice flux transiting the Robeson Channel encompasses ice floes of different ages and sizes. The typical dimensions of the ice floes examined in this study ranged from 2 to 16 km. Some ice floes were aggregates of smaller floes, which disintegrated during their journey. The 39 ice floes selected for motion tracking in the Sentinel-1 images were numbered, and the numbers
210   are used in the following analysis, although the order does not carry any significance. The tracks of 12 ice floes are shown in Fig. 4, with the floe numbers and the dates at each position attached. These ice floes were mostly heading southward, but with a few interruptions to this dominant direction. The high ice concentration in the convergent path upstream of the Robeson Channel caused reduction of the ice motion and induced meandering paths. However, in areas with less ice concentration, the ice floe motion accelerated and became more influenced by wind, as will be demonstrated in case study 1. Once the ice floes
215   crossed the bottle neck at the entrance to the Robeson Channel, they became partially relieved from the stresses induced by the surrounding ice and more responsive to other factors such as wind and current. Thus, the speed increased greatly by a factor of 1.5–5, and the drift direction followed mostly the north–south extension of the Robeson Channel, which coincided with the dominant wind direction. This direction also coincided with the dominant ocean current direction. Figure 4 also shows that the ice floes did not enter any of the fjords at the sides of the channel. In fact, many fjords become filled by locally grown
220   landfast ice early in the freezing season.

Commented [36]: This sentence has been improved.

Commented [37]: **To Reviewer 1**: These sentences are rewritten for easier understanding of readers.

Commented [38]: **To Reviewer 1**: We use specific number instead of "remarkably". All the "remarkably" is deleted from the manuscript.

[Figure]

**Figure 4**. Trajectories of 12 selected ice floes, obtained from the daily Sentinel-1 images, as they approach and pass through the Robeson Channel. Note the slow motion upstream of the channel and the faster motion through the channel. The entrance to the channel is marked by the solid line in the top left panel.

**Commented [39]: To Reviewer 2**: Added here.

225 **4.1.2 Ice floe drift speed**

Figure 5 shows the average drift speed of each ice floe (regardless of drift direction) during its entire observation time in the SAR time series, either upstream or within the Robeson Channel. Upstream of the Robeson Channel, the drift speed varied within a narrow range (4–10 km d$^{-1}$), with a typical value around 5 km d$^{-1}$. Such nearly constant drift speeds, observed under different wind speeds, suggest that the wind has a minor influence or even no influence on the ice floe drift in this area. The

230 exceptionally high average speed of ice floe #6 (~19 km d$^{-1}$) resulted when the floe drifted in a surrounding area of nearly zero ice concentration, which prevailed for three days.

**Commented [40]: To Reviewer 2**: Added here.

**Commented [41]:** Instead of "from".

The situation was different for the ice floes that drifted within the Robeson Channel. Here, the ice floe speed was much higher, typically between 14 and 45 km d$^{-1}$. One ice floe reached an extreme speed of around 99 km d$^{-1}$ on one day, as shown in case

235 study 3 below. The higher drifting speed inside the Robeson Channel can be partly explained by the low ice concentration and/or the prevalence of thin ice, particularly after the ice arch matured on 2 February 2017. Both factors gave rise to a more significant influence of wind and ocean currents on the ice drift. The large variability of the ice floe speed within the Robeson Channel, which contrasts with the nearly constant speed upstream of the Robeson Channel, can be attributed to the influence of the variable wind speed and direction. For example, the very high average speed of ice floe #4 (45 km d$^{-1}$, as shown in Fig.

240 5), which is a manifestation of the large leap in location during the period 10–13 November (Fig. 4), was instigated by a dominant southward wind between 20 and 40 km h$^{-1}$ during that period. On the other hand, the relatively slow average drift speed of ice floe #5 (15 km d$^{-1}$, as shown in Fig. 5) resulted from a reversed wind direction that blew from south to north at 10–20 km h$^{-1}$ between 18 and 26 October. This adverse wind neutralized the action of the southward current.

**Commented [42]:** This sentence has been improved.

[Figure]

245    **Figure 5.** Average speed of individual ice floes during the periods upstream and within the Robeson Channel. The last seven floes (#33–39) drifted within the Robeson Channel with highly variable speeds.

**Commented [43]: To Reviewer 2**: Instead of "floated".

**4.1.3 Driving forces of ice floe motion**

**Commented [44]: To Reviewer 1 & 2**: The whole paragraph is rewritten thoroughly and describes our exploration of tidal effects on ice drift. Due to lack of regular generation of tidal data in this area,
[revised manuscript text omitted]

**Commented [45]:** Sentences are rewritten for easier understanding.

**Commented [46]: To Reviewer 1**: The sentence structure is modified.

**Commented [47]:** Instead of "The pack ice motion is driven by".

**Commented [48]: To Reviewer 1**: These sentences are rewritten to show a possible influence of current SSH to ice drift. Detailed information can be found in the Response Letter.

[Figure]

**Figure 7.** Sequence of Sentinel-1A/1B images for an area north of the Robeson Channel. The dotted curve marks an area of consolidated ice (still visible in the 26 October image). Ice cracked in this area on 7 November and an ice arch was formed on 13 November. The star in the middle panel of the top row marks Ellesmere Island. Ice floes that made their way to the Robeson Channel originate from the west (not north), following the path shown by the arrow in the 1 December image.

*Case study 1: two floes drifting upstream of the Robeson Channel*

Figure 8 shows sequential Sentinel-1 images (26 September to 7 October 2016) of a segment just upstream of the Robeson Channel, where two ice floes appear. Ice floe #2 is marked by the grey dot, a natural low backscatter area, and floe #3 is marked by the star. The corresponding maps of the 3-hour ERA5 reanalysis wind vectors are presented in Fig. 9. The daily speeds of each ice floe are listed in Table 2, along with the wind and qualitative concentration data. This information helps in defining the impact of the wind on the ice floe drift, as explained below.

The image for 26 September shows the two ice floes surrounded by open water and thin ice. The wind between the two satellite overpasses on 26 and 27 September (averaging 33 km h$^{-1}$) was partly heading northeast or southeast. This relatively high wind,

**Commented [49]: To Reviewer 1**: The misspelled word "bath" is corrected.

**Commented [50]: To Reviewer 1**: Redundant information is deleted. And the sentence is rewritten for better expression.

**Commented [51]: To Reviewer 1**: We eliminated many of those parenthetical statements to make the reading flows better.

combined with the less resistive ice in the surrounding, caused floe #3 to drift northeast at a top speed of 24 km d$^{-1}$ (Table 2). Between 27 and 28 September, relatively light wind (<20 km h$^{-1}$) blew in the opposite direction, but floe #3 drifted southeast

325   at 18.05 km d$^{-1}$ because this path was the path of less resistance. Floe #2 did not move far with a drift speed of 2.84 km d$^{-1}$ as it was surrounded by ice. Between 28 and 29 September, the light wind did not change and the two ice floes remained at the same locations. When the wind direction switched to the southeast between 29 and 30 September, with the speed reaching 30 km h$^{-1}$, the two floes drifted in the same direction, with floe #2 reaching a speed of 15.72 km d$^{-1}$, as shown in Table 2. Here, once again, the path was nearly ice-free. Between 30 September and 1 October, when a southwestward wind blew at nearly 30

330   km h$^{-1}$, the ice drift accelerated to nearly 9 km d$^{-1}$. After 1 October, the wind abated but the drift continued in a southeast direction at a moderate speed of 2–6 km d$^{-1}$. When the northward wind exceeded 40 km h$^{-1}$ between 4–7 October, before it was reduced to less than 30 km h$^{-1}$, the ice floe drift did not follow the wind action in the first two days because the two floes were surrounded by high ice concentrations. Nevertheless, southwestward drift was observed between 4–5 October, particularly for floe #2 (Fig. 8), following the strong northwestward wind during the same period (Fig. 9). This case study

335   demonstrates the effective role of wind on ice floe drift when the floe is surrounded by thin ice or water.

[Figure]

**Figure 8.** Sequential Sentinel-1A/1B images (dates are shown) showing the advancement of two ice floes. Floe #2 is marked by a grey dot (a natural low backscatter area), and floe #3 is marked by a star. The ice concentration surrounding each floe is visible and can be qualitatively estimated.

[Figure]

**Figure 9.** Maps of the 10-m level wind vector (km h$^{-1}$) from the ERA5 reanalysis. Each panel has vectors from the four grid points surrounding the locations of ice floes #2 and #3 every 3 hours during the time between the daily overpasses of Sentinel-1. For example, the 26–27 September panel shows the 3-hour vectors that were available between the period that spans the satellite acquisition times of 26 and 27 September.

345

**Table 2.** Drift speed of ice floes #2 and #3 (shown in Fig. 8) during the period between the acquisitions of the two successive daily Sentinel-1 images. The period is shown in the first column.

| Date | Avg. speed (km d$^{-1}$) | | Wind speed (km h$^{-1}$) | Qualitative ice concent. |
|---|---|---|---|---|
| | Floe #2 | Floe #3 | | |
| 26–27 September | 9.68 | 24.17 | 33 | low |
| 27–28 September | 2.84 | 18.05 | 15 | low |
| 28–29 September | 3.97 | 3.32 | 10 | low |

**Commented [52]: To Reviewer 1**: The caption is modified to make it comprehensible.

**Commented [53]: To Reviewer 2**: Instead of "two successive Sentinel-1 coverage".

**Commented [54]: To Reviewer 1**: The unit has been corrected.

[revised manuscript text omitted]

**Commented [55]: To Reviewer 2**: Two paragraphs and the previous Figure 11 are deleted to shorten the manuscript.

**Commented [56]:** The following paragraph and Table 3 are added here using multiple linear regression to distinguish between the influence on ice drift from wind speed and current speed.

variable, in the absence of other independent variables. Once again, the results show the more significant contribution of the
wind. The variance inflation factor (VIF) should be <10, otherwise there is severe multicollinearity in the model. The
405 conclusion from these results is that wind speed has a greater influence on ice floe motion than ocean current in the open drift
(40–60% ice concentration) or very open drift (<40% ice concentration) regimes in the Robeson Channel.

**Table 3**. Results from the multivariate regression analysis showing the contributions of wind and ocean current to ice floe
motion.

| Parameter | Standardized coeff. | Statistical significance | Pearson corr. coeff. | Partial corr. coeff. | VIF |
|---|---|---|---|---|---|
| Intercept | | <0.001 | | | |
| Wind speed | 0.729 | <0.001 | 0.766 | 0.753 | 1.053 |
| Current speed | 0.165 | <0.001 | 0.328 | 0.251 | 1.053 |

410 *Case study 2: An ice floe moving in the drifted ice regime in the Robeson Channel*

Figure 11 shows a sequence of daily Sentinel-1 images from 14 to 19 November, where many ice floes originating from the
north of the Robeson Channel can be seen. The path of the ice floe marked with the asterisk (floe #29) is linked to the coincident
wind vectors in Fig. 12. This ice floe moved southward at a speed of 27.0 km d$^{-1}$ between 14 and 15 November. During this
period, the wind speed was between 5 and 20 km h$^{-1}$, with a wide range of directions, but the strongest wind blew from the
415 south to the northeast (Fig. 12). Apparently, this drift was more influenced by the north–south current in this case (daily current
maps are not shown). The momentum of the incoming ice floes from the north is a possible explanation for this southward
motion. Between 15 and 16 November, relatively strong wind with a speed between 20 and 37 km h$^{-1}$ blew from the south.
However, the entire set of floes appear to have drifted eastward. Once again, the wind did not trigger this motion. SSH can be
a possible cause because it has a gradient that matches the drift direction (Fig. 7). Between 16 and 18 November, the same
420 strong wind, which approached 30 km h$^{-1}$, continued to blow to the northeast (Fig. 12), and the entire set of floes responded
by drifting in the same direction. Floe #29 reached its highest speed of 11.77 km d$^{-1}$ on 17 November. Between 18 and 19
November, the wind diminished, but a group of ice floes appeared to swirl clockwise. The current continued to be southward,
and there was 100% local ice concentration around floe #29.

**Commented [57]: Reviewer 1**: Wekerle et al. (2013) mentioned that the SSH difference between the Arctic Ocean and Baffin Bay. Detailed information can be found in Response Letter.

**Commented [58]: To Reviewer 1**: Instead of "continues".

**Commented [59]: To Reviewer 1**: The sentence is rewritten for better expression.

[Figure]

425 **Figure 11**. A sequence of daily Sentinel-1 images showing the path of a number of ice floes. The floe marked by the asterisk (floe #29) is the subject of the comments in the text. Dates of the images are shown, as well as the speed of the marked floe.

**Commented [60]:** Since the previous Figure 11 was deleted, the subsequent ones have been renumbered.

[Figure]

430 **Figure 12.** Maps of the 10-m level wind vectors (km h$^{-1}$) from the ERA5 reanalysis. Each panel has vectors from the four grid points surrounding the location of ice floe #29 every 3 hours during the time between the daily overpasses of Sentinel-1. For example, the 14–15 November panel has the 3-hour vectors between the acquisition times of 14 and 15 November.

**Case study 3: An ice floe drifting in the polynya within the Robeson Channel**

435  Fig. 13 shows the track of ice floe #38, which broke off from landfast ice at the Greenland side and drifted north then south in the polynya regime. The trajectory covers the period from 8 to 22 February 2017, after the arch formed. The daily wind vector maps associated with the selected floe location are presented in Fig. 14. Between 10 and 11 February, northward wind dominated, although this never exceeded 20 km h$^{-1}$. The ice floe drift of nearly 12 km d$^{-1}$ matched the wind direction. Between 13 and 17 February, the northward wind accelerated, reaching 40 km h$^{-1}$ and then 50 km h$^{-1}$. The ice floe moved in the same

440  direction, with its speed reaching 11.8 km d$^{-1}$, 32.2 km d$^{-1}$ and 20.4 km d$^{-1}$ on 15, 16 and 17 February, respectively. The speed was significantly reduced to 3.7 km d$^{-1}$ on 18 February as the floe approached the ice arch. After this day, the wind blew from the north and the ice floe changed its direction of motion to advancing southward. It is interesting to note the high ice floe speed of 43.0 km d$^{-1}$ between 20 and 21 February and the highest speed of 99.1 km d$^{-1}$ between 21 and 22 February. The latter was triggered by the highest wind encountered in this study, which gusted to 50 km h$^{-1}$. However, it is important to recall that

445   the surface current drives ice motion in the same direction. This case study demonstrates that the influence of the wind on ice

motion is the greatest in areas of thin ice and water.

[Figure]

**Figure 13**. Trajectory of an ice floe (floe #38) that separated from landfast ice and drifted in the polynya regime downstream of the ice arc.
450   The track is shown from 8 to 22 February 2017.

[Figure]

**Figure 14.** Maps of the 10-m level wind (km h$^{-1}$), as shown in Fig. 12, but for ice floe #38, which was separated from landfast ice and drifted in the polynya downstream from the ice arch formed at the inlet of the Robeson Channel.

455

**4.2 Formation of the ice arch**

The ice arch phenomenon is a necessary condition for polynya formation downstream. Polynyas can be driven by wind action that removes newly formed ice (latent heat polynya), and/or warm upwelling ocean water that melts the ice as soon as it is forms (sensible heat polynya) (Smith et al., 1990). However, if the flux of ice from a nearby source continues to feed into the area that would become a polynya, then the polynya can only be formed if a natural obstacle develops to block the flux. This obstacle could be an ice arch, which is a mechanically strong formation that can withstand the massive dynamic load of the advected sea ice. Clearly, this factor is irrelevant to coastal polynyas as they are backed by land. This is more common in the Antarctic region (Nihashi et al, 2015). In the case of the Robeson Channel, an ice arch commonly forms at the inlet of the channel, blocking the ice flux from the Lincoln Sea into the Robeson Channel. The ice arch may collapse a few weeks after formation or persist as late as mid-August (Samelson et al., 2006). More historical context about the ice arches that form at the inlet of the Robeson Channel can be found in Kwok et al. (2010), Ryan and Münchow (2017) and Moore and McNeil (2018). The ice arch observed in the present dataset started its development on 24 January, matured on 1 February and collapsed in

**Commented [61]: To Reviewer 1**: Instead of "Polynya". "It is well known that" is also deleted.

**Commented [62]: To Reviewer 1**: This paper is added as the reference for Antarctic polynya statement.

**Commented [63]: To Reviewer 1**: Three papers on the context of this ice arch are listed here for readers.

[revised manuscript text omitted]